# Afforestation impact on soil temperature in regional climate model simulations over Europe

Giannis Sofiadis[1], Eleni Katragkou[1], Edouard L. Davin[2], Diana Rechid[3], Nathalie de Noblet-Ducoudre[4], Marcus Breil[5], Rita M. Cardoso[6], Peter Hoffmann[3], Lisa Jach[7], Ronny Meier[2], Priscilla A. Mooney[8], Pedro M.M. Soares[6], Susanna Strada[9], Merja H. Tölle[10], Kirsten Warrach Sagi[7]

[1]Department of Meteorology and Climatology, School of Geology, Aristotle University of Thessaloniki, Thessaloniki, Greece
[2]Department of Environmental Systems Science, ETH Zurich, Zurich, Switzerland
[3]Climate Service Center Germany (GERICS), Helmholtz-Zentrum Hereon, Fischertwiete 1, 20095 Hamburg, Germany.
[4]Laboratoire des Sciences du Climat et de l'Environnement; UMR CEA-CNRS-UVSQ, Université Paris-Saclay, Orme des Merisiers, bât 714, 91191 Gif-sur-Yvette CÉDEX, France.
[5]Institute of Meteorology and Climate Research, Karlsruche Institute of Technology, Karlsruche, Germany.
[6]Instituto Dom Luiz (IDL), Faculdade de Ciencias, Universidade de Lisboa, 1749-016 Lisboa, Portugal.
[7]Institute of Physics and Meteorology, University of Hohenheim, Stuttgart, Germany.
[8]NORCE Norwegian Research Centre Bjerknes Center for Climate Research, Bergen, Norway.
[9]International Center for Theoritical Physics (ICTP), Earth System Physics Section, Trieste, Italy.
[10]Universität Kassel, Center of Environmental Systems Research (CESR), Wilhelmshöher Allee 47, 34117 Kassel, Germany.

*Correspondence to*: Giannis Sofiadis (sofiadis@geo.auth.gr)

**Abstract.** In the context of the first phase of the Euro-CORDEX Flagship Plot Study Land Use and Climate Across Scales (LUCAS), we investigate the biophysical impact of afforestation on the seasonal cycle of soil temperature over the European continent with an ensemble of ten regional climate models. For this purpose, each ensemble member performed two idealized land cover experiments in which Europe is covered either by forests or grasslands. The multi-model mean exhibits a reduction of the annual amplitude of soil temperature (AAST) due to afforestation over all European regions, although this is not a robust feature among the models. In Mediterranean, the spread of simulated AAST response to afforestation is between -4 ºC to +2 ºC at 1 meter below the ground while in Scandinavia the inter-model spread ranges from -7 ºC to +1 ºC. We show that the large range in the simulated AAST response is due to the representation of the summertime climate processes and is largely explained by inter-model differences in leaf area index (LAI), surface albedo, cloud fraction and soil moisture, when all combined into a multiple linear regression. The changes in these drivers essentially determine the ratio between the increased radiative energy at surface (due to lower albedo in forests) and the increased sum of turbulent heat fluxes (due to mixing-facilitating characteristics of forests), and consequently decide the changes in soil heating with afforestation in each model. Finally, we pair FLUXNET sites to compare the simulated results with observation-based evidence of the impact of forest on soil temperature. In line with models, observations indicate a summer ground cooling in forested areas compared to open lands. The vast majority of models agree with the sign of the observed reduction in AAST, although with a large variation in the magnitude of changes. Overall, we aspire to emphasize the biophysical effects of afforestation on soil temperature profile with this study, given that changes in the seasonal cycle of soil temperature potentially perturb crucial biochemical processes.

Robust knowledge on biophysical impacts of afforestation on soil conditions and its feedbacks on local and regional climate is needed in support of effective land-based climate mitigation and adaption policies.

## 1 Introduction

There is currently a strong policy focus on afforestation as a possible greenhouse gases mitigation strategy to meet ambitious climate targets (Grassi et al., 2017). The biogeochemical effects of afforestation or reforestation are mostly related to increased carbon stocks stored in vegetation and soil, as the total carbon stored in forests is nearly three times larger than carbon stored in croplands (Devaraju et al., 2015). However, understanding the full climate consequences of the large-scale deployment of such a strategy requires to consider also the biophysical effects of afforestation arising from changes in evapotranspiration efficiency, rooting depths and soil water holding capacity, surface roughness and surface albedo (Betts, 2000; Bonan, 2008; Davin and de Noblet-Ducoudre, 2010; Perugini et al., 2017; Duveiller et al., 2018).

Previous studies have attempted to quantify the biophysical impact of land-use changes (LUC) on global scale, employing either an ensemble of earth system models (ESMs) (Pitman et al., 2009; Noblet-Ducoudré et al., 2012; Boisier et al., 2012; Lejeune et al., 2018) or applying a single ESM individually (Claussen et al., 2001; Davin et al., 2007; Li et al., 2016). Davin and de Noblet-Ducoudre, 2010 analysed an ESM's sensitivity to idealized global deforestation, indicating that the net biophysical impact results from the balance between radiative and non-radiative processes. In the same study, deforestation induced a warming over the tropical zone owing to a reduction in evapotranspiration rate and surface roughness, whereas a deforestation-induced cooling simulated over the temperate and boreal zones, because an albedo increase provided the dominant influence in these regions. In the context of Land-Use and Climate, IDentification of Robust Impacts model intercomparison project, Noblet-Ducoudré et al., 2012 diagnosed the LUC effects over North America and Eurasia between the present and the pre-industrial era. They found that deforestation caused a systematic surface albedo increase across all seasons, leading to a reduction in available energy accompanied by a decrease in the sum of turbulent fluxes. Furthermore, Lejeune et al., 2018 using a suite of simulations from Coupled Model Intercomparison Project Phase 5 concluded that moderate deforestation over Eurasia and North America has substantially led to a local warming of present-day hot extremes since pre-industrial time.

Regional Climate Models (RCMs) constitute dynamical downscaling techniques applied over limited-area domains with boundary conditions either from global reanalysis or global climate model (GCM) output (Katragkou et al., 2015; Giorgi, 2019; Rummukainen, 2016). RCMs operate on higher resolutions than GCMs adding value in regions with complex orography and capturing exreme events (Soares et al., 2012; Cardoso et al., 2013; Warrach-Sagi et al., 2013). RCMs have been also used individually to address the LUC effects on regional scale (Gálos et al., 2013; Tölle et al., 2018; Cherubini et al., 2018; Belušić et al., 2019). Lejeune et al., 2015 used a state-of-the-art RCM to explore the biophysical impacts of possible future deforestation on Amazonian climate. They demonstrated that the projected land cover changes for 2100 could increase the mean annual surface temperature by 0.5 °C and decrease the mean annual rainfall by -62 mm year$^{-1}$ compared to present conditions. Similar

findings were demonstrated for a deforestation scenario over South-East Asia in Tölle et al., 2017. Strandberg and Kjellström, 2019 performed regional climate simulations undertaking scenarios of maximum deforestation/reforestation over Europe using

a single RCM. They concluded that total deforestation could result in a warmer summer by 0.5 ºC - 2.5 ºC in Europe, while the effect on precipitation was less certain. A more realistic land cover change study based on convection-permitting regional climate model simulations (Prein et al., 2015) suggested that increased cultivation of bioenergy crops by poplar trees can reduce future local maximum temperatures by up to 2 °C in central Europe (Tölle and Churiulin, 2021).

The crucial need for the assessment of LUC biophysical impacts on regional scale over Europe is addressed by the Land Use

and Climate Across Scales (LUCAS) initiative (Rechid et al., 2017) which had been approved by WCRP CORDEX as a Flagship Pilot Study (FPS). It was initiated jointly by the European branch of the Coordinated Downscaling Experiments EURO-CORDEX (Jacob et al., 2014, 2020) and the global model intercomparison study "Land-Use and Climate, IDentification of robust impacts" LUCID (Noblet-Ducoudré et al., 2012). In the first phase of LUCAS, for the first time multi-model and multi-physics simulations were performed under a common experimental protocol to address the RCMs sensitivity

to idealized land use changes in Europe. The first experiment assumed a maximum forest coverage while the second a maximum grass coverage over Europe.

Contrasting these two idealized LUC experiments, Davin et al., 2020 analyzed the robustness of RCMs responses to afforestation and according to their results, afforestation implied an albedo-induced warming over northern Europe during winter and spring. Furthermore, the summer near-surface temperature response to afforestation was subject to large

uncertainty, strongly related with disagreement among models in land-atmosphere interactions. Analyzing a part of RCM ensemble established within LUCAS FPS, Breil et al., 2020 examined the impact of afforestation on the diurnal temperature cycle in summer. Their results revealed that afforestation dampened the diurnal surface temperature cycle, while the opposite was true for the temperature in the lowest atmospheric model level. Afforestation could also enhance snow melt and modify the land-atmosphere interactions in sub-polar and alpine climates through changes in snow-albedo effect in winter and spring

(Mooney et al., 2021).

The responses of atmospheric processes to afforestation have been extensively discussed in previous studies. However, the changes in soil temperature profile following the afforestation remain unexplored up to now in LUCAS community. MacDougall and Beltrami, 2017 suggested that deforestation may has led to a long-term warming of the ground, associated with a reduction of heat fluxes towards the atmosphere. Here, we investigate the biophysical impact of afforestation on soil

temperature across Europe, as simulated by a suite of ten RCMs established within the frame of the first phase of FPS LUCAS. The comparison between two extreme LUC scenarios, representing the Europe entirely covered by forest and grass respectively, enable us to gain insights into the biophysical impacts of theoretical afforestation on soil temperature variations (Sect. 3.1). In order to explain the inter-model spread in annual amplitude of soil temperature (Sect. 3.4), we examine the changes in surface energy balance components with respect to differences in land-use parameters across RCMs (Sect. 3.2) and

the response of soil moisture content to afforestation in summer (Sect. 3.3). In addition, we compare the simulated impact on AAST with observational evidence based on FLUXNET paired sites, classified as forest or open land (Sect. 3.5).

## 2 Data and Methods

### 2.1 Regional Climate Model ensemble

Two idealized LUC experiments are carried out using an ensemble of ten RCMs. **Table 1** provides a brief description of the RCM ensemble characteristics, while more information about the land and atmospheric setups can be found in Davin et al., 2020 and in **Table S1** in the supplementary material. Compared to Davin et al., 2020 the current model ensemble includes simulations from two additional RCMs (CCLM-CLM5.0 and WRFc-NoahMP) while one of the RCMs (RCA) is not included here because the necessary variables for the analysis not recorded. Compared to CCLM-CLM4.5, CCLM-CLM5.0 is coupled with a modified version of CLM 5.0 (Lawrence et al., 2019) that includes biomass heat storage (Swenson et al., 2019; Meier et al., 2019). WRFc-NoahMP shares the same land component as WRFb-NoahMP but differs in the atmospheric set-up. Namely, WRFc-NoahMP used the YSU scheme (Hong et al., 2006) as planetary boundary layer parameterization, as opposed to MYNN Level 2.5 PBL (Nakanishi and Niino, 2009) in WRFb-NoahMP In addition, new simulations were carried out for WRFb-NoahMP and WRFb-CLM4.0 to address minor bug fixes.

### 2.2 Experimental design

In LUCAS, each participating RCM undertook two different simulations, applying the same experimental design. In the first experiment, called FOREST, models are forced with a vegetation map representing a Europe fully covered by trees, where they can realistically grow. Bare lands and water bodies were conserved as in original model maps. In the second experiment, called GRASS, the models integrate the same vegetation map, with the only difference that trees are entirely replaced by grasslands. These maps are shown in **Figure S1** and detailed description about the creation of maps and the way they are implemented into the respective RCMs can be found in Davin et al., 2020. All simulations are performed over the Euro-CORDEX domain (Jacob et al., 2020) with a spatial resolution of 0.44º (~50 km), forced by ERA-Interim reanalysis data (Dee et al., 2011) at their lateral boundaries and at the lower boundary over sea. Our analysis covers the 30-year period 1986-2015 and focuses on the following eight European sub-regions as described in Christensen and Christensen, 2007: Alps , British Isles , Eastern Europe , France , Iberian Peninsula , Mediterranean , Mid-Europe  and Scandinavia (**Figure 1**).

We consider the FOREST minus GRASS differences, implying the impact of theoretical maximum afforestation on soil temperature in Europe. Fourier's second law of heat conduction is widely used by LSMs to update temperature in each soil layer (Eq. 1):

$$\frac{\partial T}{\partial t} = \frac{\partial}{\partial z}\left[k * \frac{\partial T}{\partial z}\right]$$

where $\frac{\partial T}{\partial t}$ is the time rate of soil temperature (K s$^{-1}$) and $\frac{dT}{dz}$ is the spatial gradient of soil temperature (K m$^{-1}$) in the vertical direction z (m). The quantity k represents the thermal diffusivity (m$^2$ s$^{-1}$) defined at the layer node depth z(m) and is equal to the ratio of thermal conductivity to volumetric heat capacity (p * $c_m$, where p is mass density and $c_m$ specific heat capacity per unit mass). In RCMs, k is time dependent and is parameterized depending on soil type and composition (mineral components,

organic matter content), on bulk density and soil wetness. In our experiments, soil texture remains unchanged and RCMs do not account for possible occurrence of heat sources or sinks (such as organic matter or carbon decomposition) in the realm where soil heat flow takes place. Thus, the potential changes in soil wetness with afforestation constitute the main driver of differences in soil thermal diffusivity in our experiments. In this way, we use soil moisture response to afforestation as a potentially explanatory variable of soil temperature variations in RCMs.

Similar to Breil et al., 2020, we employ the residual of energy balance at land surface in order to express the surface energy input into the ground. Specifically, we define as energy input into ground the residual energy amount resulting from available radiative energy (net shortwave + incoming longwave radiation) minus the sum of turbulent heat fluxes (latent and sensible heat flux), without accounting for likely deviation of surface energy budget from assumed balance in models (Constantinidou et al., 2020b). Our analysis on the changes of surface energy balance components due to afforestation is carried out for summer season, when models disagree both on the sign and magnitude of soil temperature response. Thus, the land surface is assumed to be snow-free. Also, the current RCMs do not account for heat storage into biomass over land surface, apart from CCLM-CLM5.0. A detailed description on the structure of land-atmosphere exchange in the different LSMs is provided in Breil et al., 2020.

### 2.3 FLUXNET observational data

We use measured or high-quality gap-filled data of soil temperature on monthly scale from the FLUXNET2015 Tier 2 dataset to complement the model-based analysis. Detailed documentation on data and processing methods can be found in Pastorello et al., 2020.

In order to extract the potential effect of afforestation from observations, we employ a space-for-time analogy by searching for pairs of neighboring flux towers located in forest (deciduous, evergreen or mixed trees) and open land (grasslands or croplands), respectively. This approach has been used in previous studies aiming to investigate biophysical impacts of local LUC and evaluate LSM performance (Broucke et al., 2015; Chen et al., 2018). In search for site pairs, the following criteria were defined: the two sites have to 1) be located in the Euro-CORDEX domain, 2) differ in the type of vegetation, one site being forested and the other one being either cropland or grassland, 3) have a linear distance within the horizontal resolution of the performed simulations (less than 50 km), 4) have a common measurement period of at least two years, and 5) provide measurements at common depth below the ground surface. In total, we found 14 sites that met our criteria and combined in ten pairs. Their locations are depicted in **Figure 1** and their characteristics are reported in **Table 2**. The median linear distance between the paired sites is 11.4 km and their median elevation difference is 125 m.

The close proximity between the flux towers of paired sites ensures almost similar atmospheric conditions, so that differences can be primarily attributed to the different vegetation cover. Applying a simple linear correlation test, the differences either in elevation or separation between the flux towers of paired sites are not the dominant factors in determining the changes in AAST (r = -0.2 and r = -0.3, respectively).

For comparison with the RCMs, we consider the observed mean monthly soil temperature differences (forest minus open land) averaged over all paired sites. This is then compared with the mean of the grid cells matching the locations of the observational pairs in the various RCMs (FOREST minus GRASS). Modelled soil temperature was linearly interpolated to the common measurement depth that is available for each pair site and averaged over the time period 2003-2014 which covers the observational time span.

Last but not least, the observational setup does not fully resemble the experimental design applied in RCM ensemble. The spatial scale of afforestation applied in models is significantly different from the small forest patches the flux towers are located in. The theoretical maximum afforestation in RCMs has the potential to induce changes in large-scale atmospheric circulation, which can create teleconnections (Swann et al., 2012) that modify the regional cloud cover (Laguë and Swann, 2016) and thus the regional climate conditions. Such feedbacks are not applied in observations, where most forest measurement locations are

located in relatively small forest patches surrounded by open land and is almost unlikely to alter the climate conditions on regional scale.

## 3. Results

### 3.1 Soil temperature response

The afforestation (FOREST minus GRASS) effect on the annual amplitude of soil temperature (AAST) at 1 meter below the

ground surface is shown in **Figure 2**. Similar figures can be found for temperature at soil depths of 2 cm, 20 cm and 50 cm in the supplementary material (**Figures S2, S3, S4**) AAST is calculated as the difference between the warmest and the coldest month of an average year (based on the 1986-2015 climatology), implying that the maximum and minimum value may occur in different months depending on regions.

A large range of AAST response is simulated across RCMs. The sign of differences in AAST does not change with depth in

almost all models across regions (**Figure S5**). Within the ensemble, the magnitude of AAST response at 1 meter below the ground varies across regions from -7.1 ℃ to 1.8 ℃. Six out of the ten simulations show a decrease in the AAST due to afforestation in most regions. Four out of these six ensemble members employ a version (4.0, 4.5 and 5.0) of the CLM land surface model (LSM), coupled with a different atmospheric model (CCLM, RegCM or WRF). Therefore, it can be suggested that the agreement in sign of changes between these simulations resides to a great extent in the choice of a similar LSM. Also,

the latter finding holds true for three out of ten ensemble members exhibiting the opposite behaviour, namely an increase in AAST mostly at deeper soils over southern and eastern Europe. These three members utilize the NoahMP LSM coupled to different WRF atmospheric model configurations (WRFa, WRFb and WRFc); WRFa shows the most intense and systematic changes in AAST with afforestation (close to 2 ℃ in several regions), while the other two configurations (WRFb and WRFc) show absolute changes less than 1 ℃ at all soil depths. Last, WRFb-CLM4.0 and REMO-iMOVE exhibit similar responses

with temperature changes ranging from -1 ℃ in southern Europe to +0.5 ℃ in Scandinavia.

It is worth noting that the differences between simulations with the same atmospheric model (WRFb) coupled to different LSMs (NoahMP and CLM) disagree in sign of changes, especially over southern Europe. This finding suggests again that the choice of the LSM drives in a great extent the sign of changes in AAST (increase/decrease), while the choice of the atmospheric model further modulates (dampens/enhances) the magnitude of the signal. Another sub-ensemble is built around the CCLM atmospheric model participating with three different LSMs (TERRA, VEG3D, CLM version 4.5 and 5.0) illustrating diverse results; CCLM-TERRA exhibits the strongest decrease in AAST with maximum changes exceeding -4 °C over many regions. The CCLM-CLM configurations provide similar responses with maximum changes up to -2 °C. The CCLM-VEG3D exhibits a distinct behaviour with small AAST increases over central Europe and large AAST decrease of more than -5 °C in northern Europe

To better understand the changes in AAST, we examine the afforestation effect (FOREST minus GRASS) on mean monthly temperature at 2 cm, 20 cm, 50 cm, and 1 meter below the ground over two European sub-regions, the Mediterranean (**Figure 3**) and Scandinavia (**Figure 4**). These two regions are selected as they are representative of southern and northern Europe, respectively, while similar figures can be found for all European subregions in the supplementary material (**Figures S6-S11**). Over the Mediterranean region almost all models respond to afforestation, with REMO-iMOVE exhibiting an almost constant temperature increase of small magnitude at all soil depths and seasons. From the remaining simulations, six out of the nine show that summer (maximum) soil temperatures are higher in the GRASS than in the FOREST experiment. All simulations included in this category include the CLM (coupled with CCLM, RegCM, WRF), TERRA and the VEG3D LSMs. The winter (minimum) soil temperatures in the same modelling systems are not considerably affected by afforestation and thus we can attribute the decrease in AAST, discussed before, exclusively to the summertime climate processes over the Mediterranean region. From the remaining simulations of the ensemble, WRFa-NoahMP and WRFb-NoahMP show the opposite behaviour with higher forest soil temperatures in summer, while the temperature response in WRFc-NoahMP is small and mixed across months. Similar to the first group of simulations, the winter soil temperature sensitivity to afforestation is small, and as a result the AAST in WRFa-NoahMP and WRFb-NoahMP modelling systems has a positive sign of change.

In Scandinavia, a large spread in soil temperature response is simulated across RCMs in summer. REMO-iMOVE together with WRF modelling systems exhibit a small constant warming in almost all seasons and soil depths, with WRFa-NoahMP showing the most intense warming of 1.5 °C in summer. The response of the rest modelling systems is mostly based on the selection of the land component, since the CCLM model coupled to TERRA, VEG3D and CLM provides largely different results. CCLM-TERRA and CCLM-VEG3D show a temperature decrease at all soil depths, with CCLM-VEG3D being the most responsive with changes up to -9 °C in uppermost soil layer. CCLM-CLM4.5 exhibits small sensitivity across seasons with a tendency for temperature decrease in summer (similar response from RegCM-CLM4.5), while in CCLM-CLM5.0 the sign of changes switches from negative in upper layers to positive in deeper layers. In winter, the soil temperature differences due to afforestation are small in the majority of simulations and with a tendency for an increase.

## 3.2 Surface energy availability

As reported in previous section, the simulated AAST response exhibits great variability during the summer season, when
models disagree both on the sign and magnitude of changes. For this reason, it is essential to examine the changes in the available energy to warm the ground across RCMs in summer.

**Figure 5** shows maps of the afforestation impact on the surface energy input into the ground in summer or the residual of surface energy balance, as defined in Section 2.2. The pattern of changes is largely heterogeneous between the models and correlates well with the spatial pattern of changes in AAST. The choice of LSM affects the magnitude of changes; different
scales of decrease are seen between the members sharing the CCLM atmospheric model, especially between CCLM-VEG3D and CCLM-TERRA in central Europe. CCLM-CLM4.5 and CCLM-CLM5.0 provide similar responses with larger changes in southern Europe (close to -10 $Wm^{-2}$). Furthermore, the choice of LSM drives the sign of changes over southern Europe between WRFb-NoahMP and WRFb-CLM4.0. The contribution of atmospheric component is mostly related to the magnitude of changes; between RegCM-CLM4.5 and CCLM-CLM4.5, the latter provides stronger response in southern and central Europe,
while between WRF-NoahMP modelling systems, WRFa-NoahMP stands out for its intense increase in surface energy input of more than 10 $Wm^{-2}$ in several regions.

The heterogeneity in the changes of surface energy availability with afforestation is largely consistent with the disagreement in the changes of AAST among RCMs. Thus, it is crucial to explore the origin of large inter-model spread in changes of surface energy balance in summer. Below, we examine the afforestation impact on the different components of surface energy balance
for each RCM over Mediterranean (**Figure 6**) and Scandinavia (**Figure 7**). Similar figures can be found for the rest European sub-regions in the supplementary material (**Figures S12-S17**). The analysis of differences in surface energy balance components is performed with respect to changes in land-use characteristics in each RCM, such as leaf area index (LAI), surface roughness and surface albedo. Positive (negative) values indicate an increase (decrease) due to afforestation.

In both regions, all models (except from CCLM-TERRA) consistently show an increase in net shortwave radiation at the
250 surface due to afforestation, which is a result of lower albedo in FOREST compared to GRASS experiment. The changes vary across RCMs from -5 W $m^{-2}$ to 25 W $m^{-2}$ over Mediterranean and from -15 W $m^{-2}$ to 35 W $m^{-2}$ over Scandinavia. In Scandinavia, the changes in net shortwave radiation are stronger than Mediterranean. This is attributed to the fact that the forests in Scandinavia consist of needleleaf trees, which have lower albedo values compared to broadleaf trees dominating in the rest regions of Europe. Furthermore, the WRF configurations exhibit more pronounced increases in net shortwave radiation
with respect to other RCMs, which is linked to stronger reductions in albedo values in these simulations (**Figure 6f**, **Figure 7f**). Moreover, the albedo effect is further intensified by a reduction in cloud fraction with afforestation over Scandinavia in WRF configurations (**Figure 7c**). In CCLM-TERRA, the reduced net shortwave radiation is due to a pronounced increase in cloud fraction with afforestation triggered by a strong and widespread increase in evaporation rates (Davin et al., 2020). Cloud fraction is also increased with afforestation in other CCLM configurations, however the reduced incoming shortwave radiation
is offset by the albedo effect and thus the changes in net shortwave radiation have positive sign in these simulations.

The increase in available radiative energy at the surface with afforestation is followed by an increase in sensible heat flux, which is another robust feature among simulations. According to Breil et al., 2020, the increase in sensible heat flux with afforestation is attributed to higher surface roughness values in forests compared to grasslands. Generally, the high surface roughness values favour the mixing of atmosphere and enhance the heat exchange between the surface and the upper air. In the current model ensemble, the changes in sensible heat vary across RCMs from +5 W m$^{-2}$ to +26 W m$^{-2}$ over Mediterranean and from -16 W m$^{-2}$ to +35 W m$^{-2}$ over Scandinavia. Again, the only RCM which exhibits a reduction in sensible heat flux is CCLM-TERRA over Scandinavia, because of the pronounced increase in latent heat with afforestation. Moreover, WRF configurations exhibit the strongest changes in sensible heat flux within ensemble, especially over Scandinavia. As previously shown, afforestation induced intense increase in net shortwave radiation in these simulations, owing to strong reductions in albedo in combination with decreases in cloud fraction. Thus, a larger part of radiative energy is available to be transformed into sensible heat flux in these simulations. At the same time, the high surface roughness of needleleaf trees dominating in Scandinavia facilitate the energy exchange between ground and atmosphere in the form of turbulent heat fluxes.

While RCMs consistently show an increase in sensible heat flux, the agreement is much lower for the response of latent heat flux to afforestation. In Scandinavia, a tendency for increase in latent heat is noted, but in Mediterranean the simulated response is mixed. In general, the sum of turbulent heat fluxes is increased with afforestation in all models and it's largely attributed to intense and widespread increase in sensible heat flux.

To sum up, all RCMs respond to afforestation in the same way. That is, afforestation leads to increased available radiative energy at the surface due to lower albedo values in FOREST experiment compared to GRASS. In parallel, a large part of this additional radiative energy is transformed into turbulent heat energy due to the mixing-facilitating forest characteristics, such as the high LAI and roughness values, which enhance the heat exchange between the ground and upper atmosphere. The balance between the increased available radiative energy and the increased sum of turbulent heat fluxes will determine if the surface energy input into the soil will be increased or decreased with afforestation in each RCM. Since these processes are differently weighted in each modelling system depending on land-use characteristics, the resulting energy input into the soil varies within the model ensemble in terms of the sign and magnitude of changes. In CCLM-TERRA, CCLM-VEG3D, CCLM-CLM4.5, CCLM-CLM5.0 and RegCM-CLM4.5 the soil heating is decreased with afforestation in summer over Mediterranean and Scandinavia, because the increased available radiative energy is compensated by the increased sum of turbulent heat fluxes. On the other hand, REMO-iMOVE and the sub-ensemble built around NoahMP exhibit an increase in soil heating with afforestation, since the increase in the sum of turbulent heat fluxes is not enough to compensate their pronounced increase in net shortwave radiation.

## 3.3 Soil moisture

The changes in soil moisture could also have key role in describing the simulated soil temperature response to afforestation, because they affect the thermal diffusivity within the soil column. It is expected that a drier (wetter) soil column would lead

to a larger (smaller) AAST owing to its smaller (larger) heat capacity, when considering equal soil heat fluxes between the two experiments.

In **Figure 8** we map the mean summer differences in soil moisture content (SMC) of the top 1 meter of the soil over the domain of interest (FOREST minus GRASS). A widespread soil moisture decrease is simulated over the biggest part of the domain, although with considerable variation in the magnitude of changes among the models. The choice of LSM produces a large spread of responses; within the sub-ensemble around CCLM the SMC change ranges from small decrease in CCLM-CLM4.5 and CCLM-CLM5.0 to more than -30 kg m$^{-2}$ for CCLM-TERRA in several regions. Differences in the magnitude of changes

are also present between WRFb-NoahMP and WRFb-CLM4.0. The atmospheric processes also affect the magnitude of afforestation effect on SMC; among the modelling systems sharing NoahMP, WRFa-NoahMP appears to be the most responsive, with changes exceeding -20 kg m$^{-2}$ in southern Europe. Further, many grid-cells over central and northern Europe exhibit SMC increase in WRFb-NoahMP and WRFc-NoahMP configurations, in contradiction to the extensive soil moisture reduction in WRFa-NoahMP.

The surface water balance (P-E), defined as the difference between precipitation (P) and total evapotranspiration (E), decreases with afforestation during summer in the majority of models over the whole Europe (**Figure S18**). In most simulations, the decrease in the terrestrial water budget originates from increased evapotranspiration rates with afforestation. In summer, high LAI values do not allow solar radiation to reach the ground surface, as a result soil evaporation is limited and transpiration dominates overall evapotranspiration (Bonan, 2008). Specific characteristics, such as the big leaf area, the deep roots, the great

available energy due to low albedo and the mixing of the upper atmospheric boundary layer because of the high surface roughness, enhance the transpiration rate in forests. Although, CCLM-VEG3D and WRFa-NoahMP show positive sign of changes in water balance in regions of central and southern Europe, owing to decreased evapotranspiration with afforestation. This is probably linked with low atmospheric demands for hydrates in FOREST experiment of CCLM-VEG3D (Breil et al., 2021). In WRFa-NoahMP, the use of Grell-Freitas as convection scheme, exploits the transpiration facilitating features of

forests causing extreme soil drying from very early in summer. Therefore, the evapotranspiration rate lowers with afforestation, because the dry soil is not able to satisfy the atmospheric needs for hydrates.

The soil moisture changes with depth would indirectly reveal the afforestation effect on the evapotranspiration process during summer. The water uptake for transpiration occurs in different depths within the soil column for grasslands and forests. In grasslands, the soil water needed for transpiration is extracted from shallow layers, because the large fraction of their roots is

320 located there, depleting the moisture of upper soil. On the other hand, forests have a deeper root distribution, thus consuming water from a bigger soil water reservoir. In **Figure 9** we show the afforestation-induced soil moisture changes within the top 1 meter of the soil over Mediterranean and Scandinavia. Similar plots for the other sub-regions can be found in **Figure S19** of the supplementary material. The heterogeneity of SMC changes with depth is evident in most models, especially in Mediterranean. In Scandinavia, distinct drying of the uppermost soil layers is shown by some models, especially CCLM-

325 CLM4.5 and CCLM-CLM5.0, which is related to changes in water amounts from snow melt. The different structures of land models and the various descriptions of physiological characteristics of plants in LSMs, such as the root distributions,

differentiate the pattern of SMC changes with depth among the simulations. Also, possible biases in the representation of surface fluxes potentially affect the afforestation effect on soil moisture. For example, in CCLM-TERRA the latent heat fluxes are strongly increased with afforestation, as discussed in previous studies (Davin et al., 2020; Breil et al., 2020), inducing intense drying of the soil column.

## 3.4 The origin of inter-model spread in AAST

The widespread and homogeneous soil drying with afforestation, mentioned in previous section, is not consistent with the mixed AAST response. On the other hand, it is noted higher agreement between the pattern of changes in soil heating and in AAST. In section 3.2, we showed that the afforestation impact on radiative processes, such as the decrease in surface albedo, increase the available radiative energy at the surface. In parallel, the afforestation effect on non-radiative processes, removes a large part of thermal energy from surface to atmosphere in the form of sensible heat flux. The balance between these processes will determine if the surface energy input into the soil will be increased or decreased with afforestation in each RCM. However, the above biophysical processes are differently weighted across RCMs depending on land-use characteristics, like surface roughness, albedo and LAI, which affect the turbulent mixing and the amount of the absorbed solar energy at the surface. Furthermore, the response of cloud fraction to afforestation is another important factor, which affects the soil heating, because of its impact on the incoming shortwave radiation at the surface.

With the aim to quantify the effect of changes in above mentioned quantities on the simulated AAST response to afforestation, we conduct a linear regression analysis over all the European sub-regions. More specifically, we use the mean summer changes in albedo, LAI, cloud fraction and soil moisture content as explanatory (independent) variables, to determine to what extent they influence the changes in AAST (dependent variable). When we regress all the explanatory variables against the simulated AAST response, we find that the coefficient of multiple determination ($R^2$) is above 80% in all regions, which indicates the key role of the selected drivers in shaping the effect of afforestation on soil temperature (**Figure 10**). In southern regions, Mediterranean and Iberian Peninsula, the albedo effect predicts the largest part of the inter-model spread in AAST response. Over regions of central Europe (Mid-Europe, eastern Europe, France, British Isles) the predictive ability of albedo changes remains strong, although the cloud fraction is the dominating factor which effectively explains the inter-model variance over these regions. Soil moisture also contributes to the explanation of the inter-model spread in AAST over the regions of central Europe, although is not a dominating driver. In Scandinavia, the simulated AAST response is largely explained by differences in LAI across RCMs, with cloud fraction also substantially contributing to the prediction of the inter-model spread. The changes in LAI are potentially connected with the simulated cloud fraction response, since higher LAI values could facilitate the evaporation rates triggering an increase in cloud cover. This interaction effect between two or more physical processes which are used as explanatory variables constitutes a caveat of the used statistical approach, with result to reducing the effectiveness of the corresponding drivers in predicting the response of the dependent variable.

### 3.5 FLUXNET paired sites

In this section, we compare the simulated impact on AAST with observational evidence of afforestation effect on soil temperature, based on ten FLUXNET paired sites. In winter, simulations and observations illustrate insignificant changes in soil temperature with afforestation (**Figure 11**). The magnitude of afforestation effect in the observations is amplified during summer, revealing a strong cooling up to -3 ℃. The majority of models captures the seasonal pattern of changes in soil temperature and particularly the observed summer cooling, albeit with considerable variation in the magnitude of changes. CCLM-TERRA shows the largest changes in summer soil temperature (-5 ℃), whereas WRFb-NoahMP and WRFc-NoahMP exhibit subtle summer cooling smaller than -1 ℃. On the other hand, WRFa-NoahMP, CCLM-VEG3D and REMO-iMOVE do not capture the observed signal of changes in summer, simulating a warming. Especially REMO-iMOVE shows a yearly warming, opposite to the observed cooling throughout the year. According to the observations, afforestation dampens the mean annual soil temperature range by almost -3 ℃ which is qualitatively consistent with most RCMs, in which the decrease ranges from -5 ℃ for CCLM-TERRA to -0.2 ℃ for REMO-iMOVE. Notable exception is WRFa-NoahMP which exhibits a distinct increase greater than 1 ℃ in contradiction to the observational evidence. Within the sub-ensemble of CCLM model, the selection of CLM (4.5 or 5.0) as the land component, refines the simulated impact of afforestation on AAST. Also, between the simulations sharing the same WRF atmospheric configuration (WRFb), the selection of CLM4.0 against NoahMP improves the representation of soil temperature response to afforestation.

### 4. Discussion & Conclusions

In this study, we employed the experimental design established within LUCAS FPS, to investigate the afforestation impact on soil temperature over the Euro-CORDEX domain. Two idealized land cover change experiments performed by an ensemble of ten RCMs, in which the European land surface is represented as fully covered by forest and grass, respectively. The majority of simulations showed a dampening of the annual soil temperature cycle with afforestation, owing to changes in summer soil temperature. A large inter-model spread produced, ranging from -7 ℃ to +2 ℃ depending on model and region.

The changes in AAST with afforestation found to be consistent with summer changes in available energy to warm the ground across models and regions. In other words, RCMs which showed a ground cooling following afforestation, tend to simulate a reduction in surface energy input into the ground, and vice versa. What differentiates the sign of changes in soil heating across models, is the balance between two biophysical processes, which are greatly affected by afforestation. First, it is the increased available radiative energy at the surface, due to lower albedo in forests, and second it is the increased sum of turbulent heat fluxes (mostly sensible heat flux), owing to mixing-facilitating characteristics in forests, such as high LAI and surface roughness values, which enhance the heat exchange between ground and atmosphere. Although these physical processes are differently weighted in LSMs depending on land-use characteristics, such as surface albedo, surface roughness and LAI, while subsequent atmospheric feedbacks, such as the cloud cover changes, can influence the surface fluxes. Thus, the magnitude of afforestation effect on net shortwave radiation and on turbulent heat fluxes is differently pronounced across models. In six out

of ten RCMs of the ensemble, the increased available radiative energy is compensated by the increased sum of turbulent heat fluxes, thus simulating a decrease in soil heating with afforestation and finally a reduction in soil temperature, while the opposite is true for the other four modelling systems. Finally, the changes in albedo, LAI, cloud fraction and soil moisture found to explain more than 80% of inter-model variance in AAST response in all sub-regions.

Previous studies which addressed the effects of LUC on soil temperature have reported similar results with the present work.

Ni et al., 2019 employed field monitoring on a landscape consisted of tree and grass covered ground, to investigate the soil temperature effects on root water uptake for a time period from July to November. They found that soil temperature under the grass-covered ground had larger fluctuations and slightly higher values compared to tree-covered ground in summer. Lozano-Parra et al., 2018 studied the combined effect of soil moisture and vegetation cover on soil temperature over three dryland areas of the Iberian Peninsula for two hydrological years. Under dry conditions, they found smaller daily amplitudes of soil

temperature below the tree canopies than in grasslands. Longobardi et al., 2016 used a global climate model to investigate the climate sensitivity to various rates of deforestation across the globe. According to their results, deforestation warmed the soils of the mid latitudes, because of a reduction in sensible heat fluxes that offset the induced albedo increase. Lastly, MacDougall and Beltrami, 2017 conducted a GCM experiment to study the historical deforestation impact on subsurface temperatures on global scale. They found that a soil temperature increase remains present for centuries following the deforestation, originated

from the reduction of surface energy fluxes towards the atmosphere.

In line with recent findings from observations and model-based studies (Jia et al., 2017; Ren et al., 2018; Zhang et al., 2018; Li et al., 2018), we found that afforestation induced a widespread soil moisture reduction in summer, implying smaller soil heat capacity. This was also a robust feature among the models, albeit with a considerable range in the magnitude of changes. Soil moisture decrease with afforestation resulted from large drying of deep layers, related to the fact that forests and grasslands

extract soil water for transpiration process from different soil depths. Although, the homogeneous soil drying and thus the smaller soil heat capacity is not consistent with the afforestation-induced decrease of soil temperature in the majority of models, explaining only a small part of inter-model variance in AAST response in regions of central Europe.

Based on paired observations from FLUXNET dataset, we evaluated the simulated soil temperature response to afforestation. The vast majority of models agreed with the observational evidence that showed a summer ground cooling in forested areas

compared to open land. The paired sites exhibited a mean reduction of -3 ℃ in AAST, while the simulated response varied from -5 ℃ to 1 ℃.

The current ensemble enables us to address the role of atmospheric and land processes in the representation of biophysical forcing of land cover change, since it involves simulations which share the same atmospheric model coupled to different land components, or share the same LSM with different atmospheric set-ups. The switch from CCLM to RegCM when both coupled

to CLM4.5 did not induce important changes in model results, implying the dominance of land processes in these simulations. Among the suite of models which share the NoahMP LSM, the atmospheric configuration selected for WRFb-NoahMP and WRFc-NoahMP significantly refined the afforestation effect on soil temperature, compared to WRFa-NoahMP. Future studies should focus on the evaluation of model performances, similar to Katragkou et al., 2015 and Constantinidou et al., 2020a, in

order to identify the origins of systematic biases and improve the representation of climate processes in simulations. Moreover,

our results stress the crucial role of LSM in the simulation of the biophysical effects of afforestation on soil conditions. Among the LSMs coupled to the CCLM model, the choice of CLM significantly improves the representation of afforestation impact on AAST. Also, WRF coupled to CLM4.0 agreed better with observations than WRF coupled to NoahMP. Another issue is the problematic behaviors in model performances stemming from unrealistic descriptions of the physical plant functioning in LSMs. Meier et al., 2018 improved the representation of the evapotranspiration with land cover change in CLM4.5, modifying

parameters related to transpiration process, such as the root distribution and water uptake formulation.

Research has accounted for the contribution of historical deforestation to present climate conditions. Last years, governments and non-governmental organizations are planning re/afforestation programs around the world with the purpose to mitigate the negative effects of anthropogenic activities on climate. With our study, we aspire to contribute to the deeper understanding of the scientific community on the biophysical effects of afforestation on soil conditions. Future studies focused on the

435 consequences of afforestation from biological or chemical aspect, are encouraged to consider our results, in order to draw comprehensive conclusions on important climate processes in which afforestation is involved, such as the carbon sequestration and microbial respiration.

**Code and data availability**

We used soil temperature data from the FLUXNET2015 Tier Two dataset, which can be accessed at the website

(https://fluxnet.org/)(last access: 05 March 2021, (Pastorello et al., 2020)). Simulations were forced by the ERA-Interim reanalysis data set (https://www.ecmwf.int/en/forecasts/datasets/reanalysis-datasets/era-interim) (last access:08 March 2021, (Dee et al., 2011). The source code of the Weather Research and Forecasting Model (WRF) is available by UCAR/NCAR and can be accessed at https://www.mmm.ucar.edu/weather-research-and-forecasting-model (last access: 08 March 2021, (Skamarock et al., 2008)). The documentation of COSMO-Model is available at the following link

(https://www.dwd.de/EN/ourservices/cosmo_documentation/cosmo_documentation.html), although a license is required for access (http://www.cosmo-model.org/content/consortium/licencing.htm). RegCM4 model is distributed from https://github.com/ictp-esp/RegCM (last access: 08 March 2021, (Giorgi et al., 2012)). The source code of REMO model is available on request from the Climate Service Center Germany (contact@remo-rcm.de) (Wilhelm et al., 2014). All the scripts and data upon which this study is based can be accessed at the link: 10.5281/zenodo.4588724 .

**Author contributions**

GS, EK and ELD designed the research. GS, EK, ELD, RM, DR, MB, RMC, PH, LJ, PM, PMMS, SS, MHT and KWS performed the RCM simulations. GS analyzed the data and wrote the paper with inputs from all coauthors.

**Competing interests.**

The authors declare that they have no conflict of interest.

**Acknowledgements**

The author gratefully acknowledges the Swiss Confederation for financial support through Government Excellence Scholarship for the academic year 2019-2020. The author thanks Prof. Sonia I. Seneviratne for the fruitful discussions on the progress of this study. The work of GS and EK was supported by computational time granted from the National Infrastructures for Research and Technology S.A. (GRNET S.A.) in the National HPC facility - ARIS - under project ID pr005025 and 460 pr007033_thin. ELD and RM acknowledge support from the Swiss National Science Foundation (SNSF) through the CLIMPULSE project and thanks the Swiss National Supercomputing Centre (CSCS) for providing computing resources. Rita M. Cardoso and Pedro M. M. Soares acknowledge the projects LEADING (PTDC/CTA-MET/28914/2017) and FCT-UID/GEO/50019/2019 – Instituto Dom Luiz. Peter Hoffmann is funded by the Climate Service Center Germany (GERICS) of the Helmholtz-Zentrum Hereon in the frame of the HICSS (Helmholtz-Institut Climate Service Science) project LANDMATE. 465 Lisa L. Jach, and Kirsten Warrach-Sagi acknowledge support by the state of Baden-Württemberg through bwHPC and thank the Anton and Petra Ehrmann-Stiftung Research Training Group "Water-People-Agriculture" for financial support. Susanna Strada has been supported by the TALENTS3 Fellowship Programme (FP code 1718349004) funded by the autonomous region Friuli Venezia Giulia via the European Social Fund (Operative Regional Programme 2014–2020) and administered by the AREA Science Park (Padriciano, Italy). Merja H. Tölle thanks the German Climate Computing Center (DKRZ) for providing 470 computing resources, the CLM-community for support and acknowledges the funding of the German Research Foundation (DFG) through grant 401857120. The authors gratefully acknowledge the WCRP CORDEX Flagship Pilot Study LUCAS "Land use and Climate Across Scales", and the research data exchange infrastructure and services provided by the Jülich Supercomputing Centre, Germany, as part of the Helmholtz Data Federation initiative.

**Financial support**

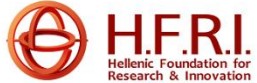

The research work was supported by the Hellenic Foundation for Research and Innovation (HFRI) under the HFRI PhD Fellowship grant (Fellowship Number: 1359).

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

**Table 1: Characteristics of the RCMs participating in the study. JLU – Justus-Liebig-Universität Gießen; BTU: Brandenburgische Technische Universität; KIT – Karlsruhe Institute of Technology; ETH – Eidgenössische Technische Hochschule Zürich; SMHI – Swedish Meteorological and Hydrological Institute; ICTP – International Centre for Theoretical Physics; GERICS – Climate Service Center Germany; IDL – Instituto Amaro Da Costa; UHOH – University of Hohenheim; BCCR – Bjerknes Center for**

**Climate Research; AUTH –Aristotle University of Thessaloniki. The full table including the parameterization schemes and settings used, can be found in Davin et al., 2020 and in Table S1 in the supplementary material.**

| Model label | Institute | RCM version | LSM | Soil column |
|---|---|---|---|---|
| CCLM-TERRA | JLU/BTU/CMCC | COSMO_5.0_clm9 | TERRA-ML (Schrodin and Heise, 2001) | 10 layers down to 15.3 m. First 9 (8) layers are thermally (hydrologically) active. The computation of soil thermal conductivity and heat capacity is described in Doms et al., 2013. |
| CCLM-VEG3D | KIT | COSMO_5.0_clm9 | VEG3D (Breil et al., 2018) | 10 layers down to 15 m. First 9 (8) layers are thermally (hydrologically) active. Soil thermal conductivity is based on Johansen, 1977 and the heat capacity on de Vries, 1963. |
| CCLM-CLM4.5 | ETH | COSMO_5.0_clm9 | CLM4.5 (Oleson et al., 2013) | 15 thermally active layers down to 42 m. The first 10 layers are hydrologically active. Soil thermal conductivity is computed according to Farouki (1981). Volumetric heat capacity is computed according to de Vries, 1963. |
| CCLM-CLM5.0 | ETH | COSMO_5.0_clm9 | CLM5.0 (Lawrence et al., 2019) | 25 thermally active layers down to 50 m. The first 20 layers are hydrologically active. Soil thermal conductivity is computed according to Farouki (1981). Volumetric heat capacity is computed according to de Vries, 1963. |
| RegCM-CLM4.5 | ICTP | RegCM4.6.1 | CLM4.5 (Oleson et al., 2013) | 15 thermally active layers down to 42 m. The first 10 layers are hydrologically active. Soil thermal conductivity is computed according to Farouki (1981). Volumetric heat capacity is computed according to de Vries, 1963. |
| REMO-iMOVE | GERICS | REMO2009 | iMOVE (Wilhelm et al., 2014) | 5 thermally active layers down to 9.8 m. One water bucket. The dependency of thermal conductivity and heat capacity on soil moisture is modelled according to Semmler, 2002. |

| | | | | |
|---|---|---|---|---|
| WRFa-NoahMP | IDL | WRF381 | NoahMP | 4 layers down to 2 m. The total heat capacity and thermal conductivity of the mineral soil are computed as proposed by Peters-Lidard et al., 1998 |
| WRFb-NoahMP | UHOH | WRF381 | NoahMP | 4 layers down to 2 m. The total heat capacity and thermal conductivity of the mineral soil are computed as proposed by Peters-Lidard et al., 1998 |
| WRFc-NoahMP | BCCR | WRF381 | NoahMP | 4 layers down to 2 m. The total heat capacity and thermal conductivity of the mineral soil are computed as proposed by Peters-Lidard et al., 1998 |
| WRFb-CLM4.0 | AUTH | WRF381 | CLM4.0 (Oleson et al., 2010) | 10 thermally and hydrologically active layers down to 3.43 m. Soil thermal conductivity is computed according to Farouki (1981). Volumetric heat capacity is computed according to de Vries, 1963. |

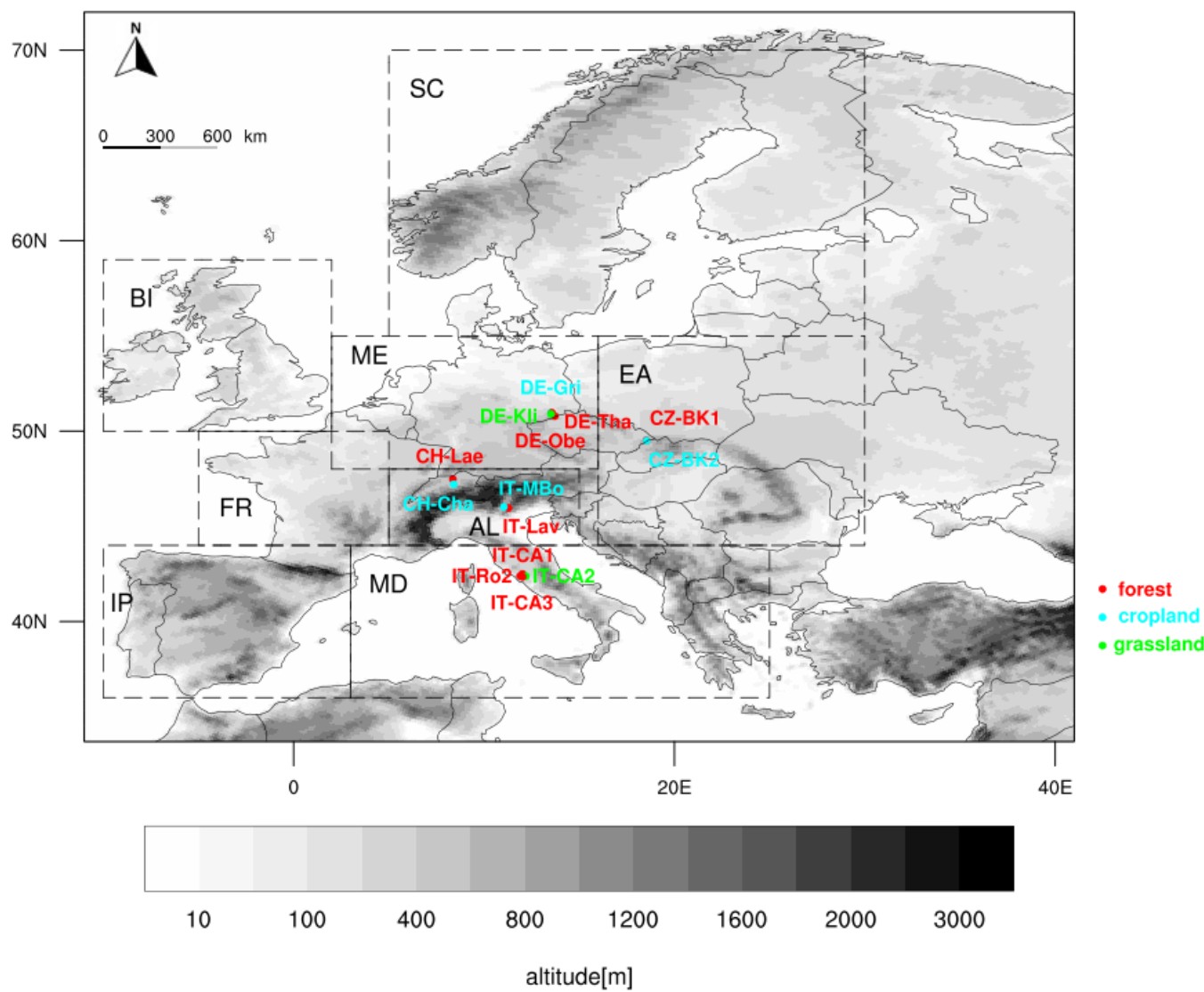

**Figure 1: Topography of the model domain and location of the observational pairs. The outlined boxes with a dashed line correspond to the eight regions on which our analysis has been focused: AL (Alps), BI (British Isles), EA (Eastern Europe), FR (France), IP (Iberian Peninsula), MD (Mediterranean), ME (Mid-Europe), SC (Scandinavia).**

**Table 2: Characteristics of the sites selected from FLUXNET2015 dataset. DBF – Deciduous Broadleaf Forest; ENF – Evergreen Needleleaf Forest; MF – Mixed Forest; CRO – cropland; GRA – grassland, as described by the International Geosphere-Biosphere Programme (IGBP) classification scheme.**

| Pair ID | FLUXNET site ID | (Latitude, Longitude) | Elevation (m) | Land cover type | Distance (km) | Time period | Measurement depth |
|---|---|---|---|---|---|---|---|
| 1 | IT-CA1 | (42.380,12.026) | 200 | DBF | 0.3 | 2011-2014 | 15cm |
| | IT-CA2 | (42.377,12.026) | 200 | CRO | | | |
| 2 | IT-CA3 | (42.380,12.022) | 197 | DBF | 0.4 | 2011-2014 | 15cm |
| | IT-CA2 | (42.377,12.026) | 200 | CRO | | | |
| 3 | IT-Ro2 | (42.390,11.920) | 160 | DBF | 8.7 | 2011-2012 | 15cm |
| | IT-CA2 | (42.377,12.026) | 200 | CRO | | | |
| 4 | CZ-BK1 | (49.502,18.536) | 875 | ENF | 0.9 | 2004-2012 | 5cm |
| | CZ-BK2 | (49.494,18.542) | 855 | GRA | | | |
| 5 | DE-Tha | (50.962,13.565) | 385 | ENF | 4.1 | 2004-2014 | 10cm |
| | DE-Gri | (50.950,13.512) | 385 | GRA | | | |
| 6 | DE-Obe | (50.786,13.721) | 734 | ENF | 23.4 | 2008-2014 | 10cm |
| | DE-Gri | (50.950,13.512) | 385 | GRA | | | |
| 7 | DE-Tha | (50.962,13.565) | 385 | ENF | 8.4 | 2004-2014 | 10cm |
| | DE-Kli | (50.893,13.522) | 478 | CRO | | | |
| 8 | DE-Obe | (50.786,13.721) | 734 | ENF | 18.4 | 2008-2014 | 10cm |
| | DE-Kli | (50.893,13.522) | 478 | CRO | | | |
| 9 | IT-Lav | (45.956,11.281) | 1353 | ENF | 19.3 | 2003-2013 | 10cm |
| | IT-Mbo | (46.014,11.045) | 1550 | GRA | | | |
| 10 | CH-Lae | (47.478,8.364) | 689 | MF | 30 | 2005-2014 | 10cm |
| | CH-Cha | (47.210,8.41) | 393 | GRA | | | |

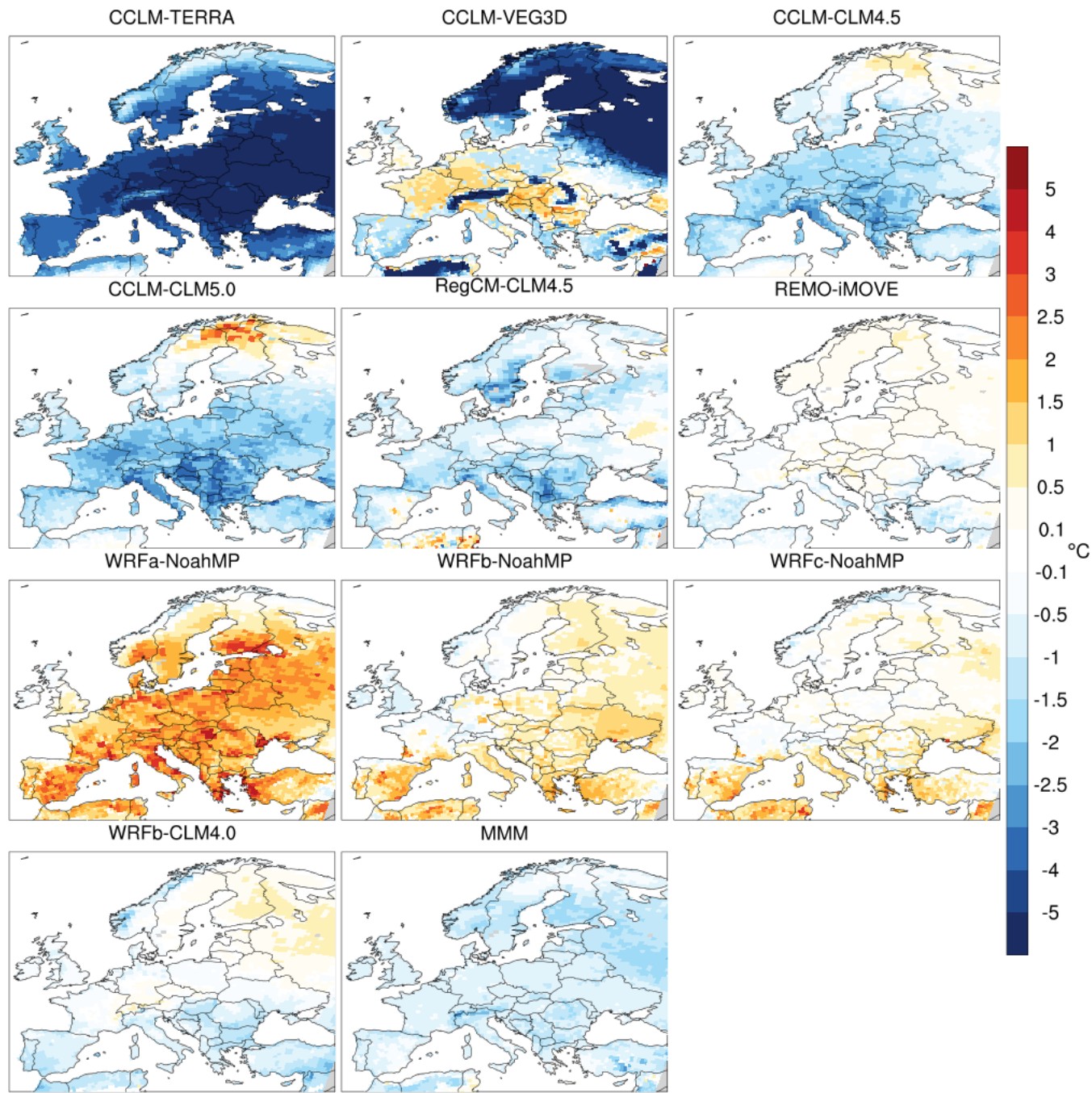

**Figure 2: Afforestation (FOREST minus GRASS) impact on the annual amplitude of soil temperature (AAST) at 1 meter depth. MMM: Multi-Model-Mean. Positive (negative) values mean increase (decrease) with afforestation.**

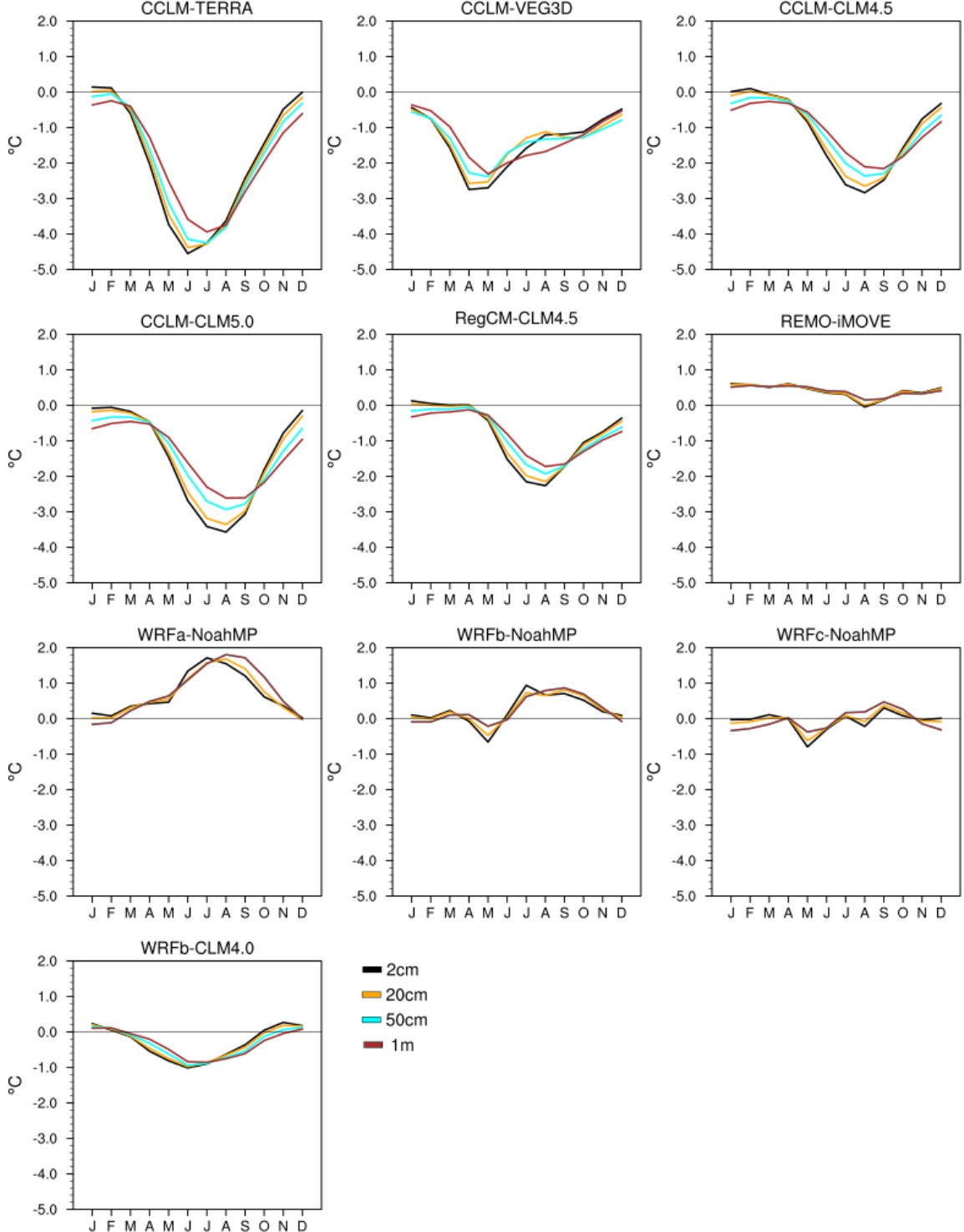

**Figure 3: Afforestation impact (FOREST minus GRASS) on mean monthly soil temperature at four different soil depths over Mediterranean.**

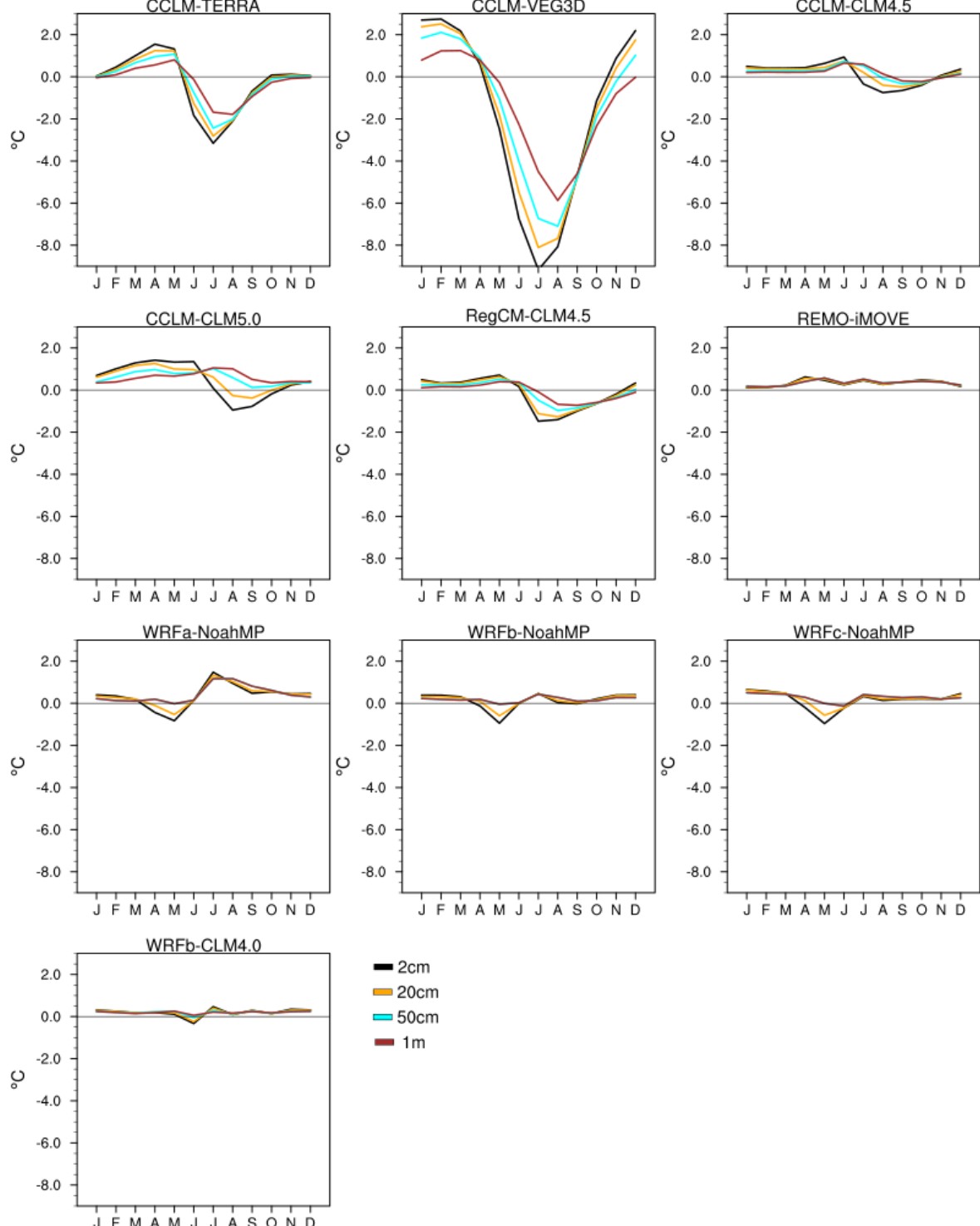

**Figure 4: Afforestation impact (FOREST minus GRASS) on mean monthly soil temperature at four different soil depths over Scandinavia.**

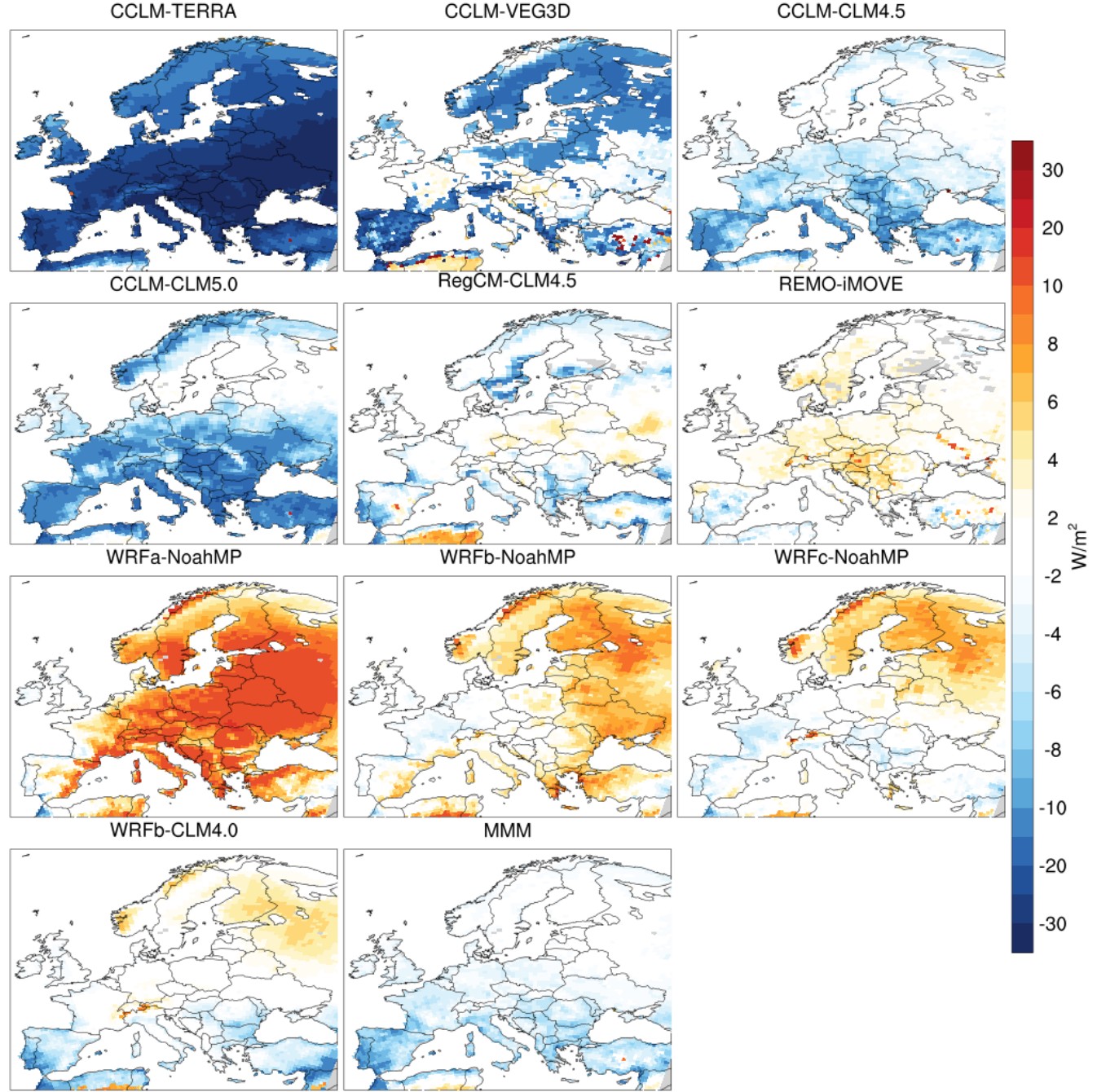

**Figure 5: Afforestation impact (FOREST minus GRASS) on the surface energy input into the ground (W m$^{-2}$) during summer.** Positive (negative) values mean increase (decrease) with afforestation.

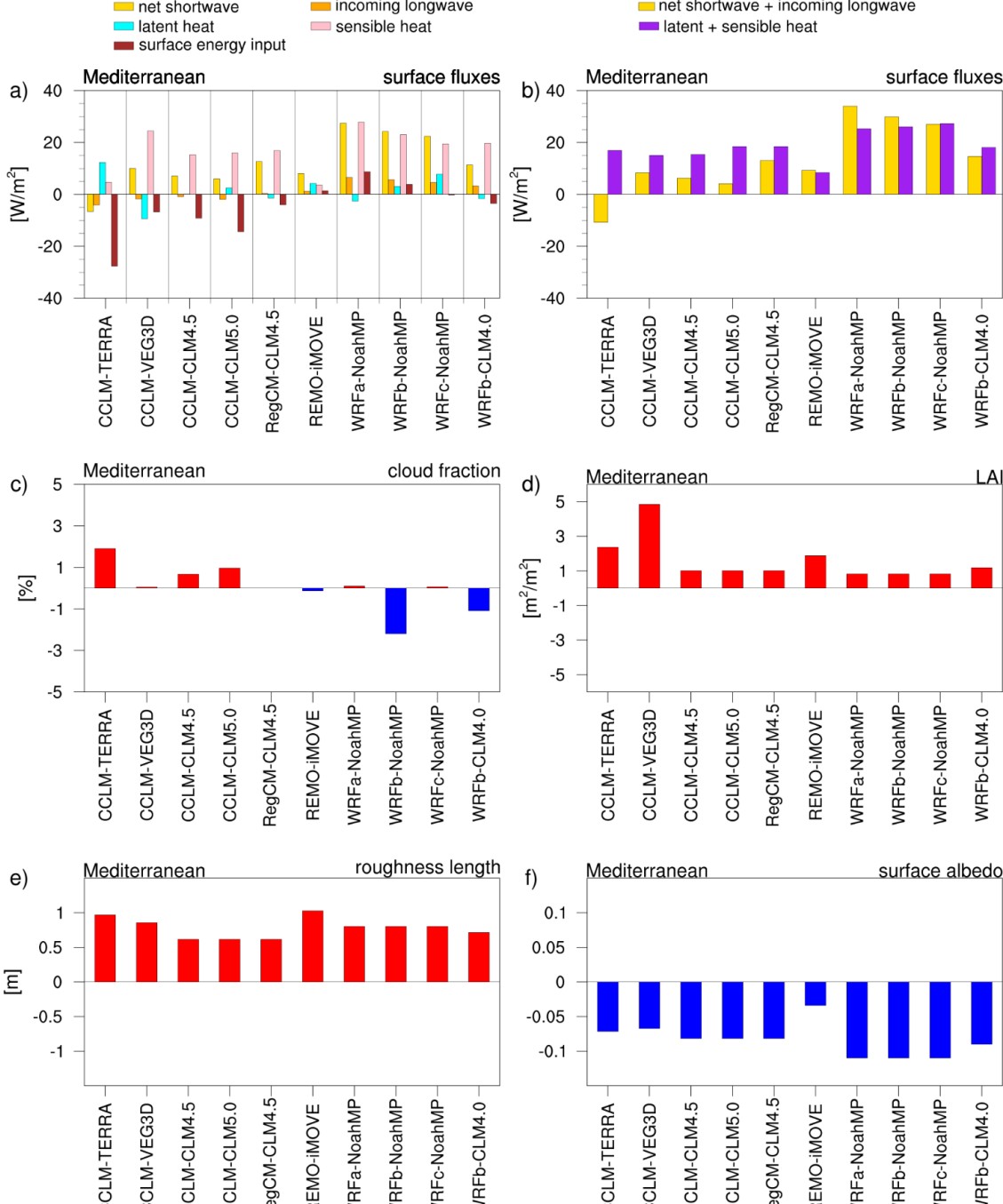

**Figure 6: (a)** Changes in surface energy balance components (FOREST minus GRASS) averaged over Mediterranean in summer, **(b)** The changes in available radiative energy at the surface and in the sum of turbulent heat fluxes with afforestation (FOREST minus GRASS), **(c)** Cloud fraction response to afforestation across models, and the inter-model differences in leaf area index (LAI) **(d)**, surface roughness **(e)** and surface albedo **(f)** in summer (yearly maximum). Positive (negative) values mean increase (decrease) with afforestation.

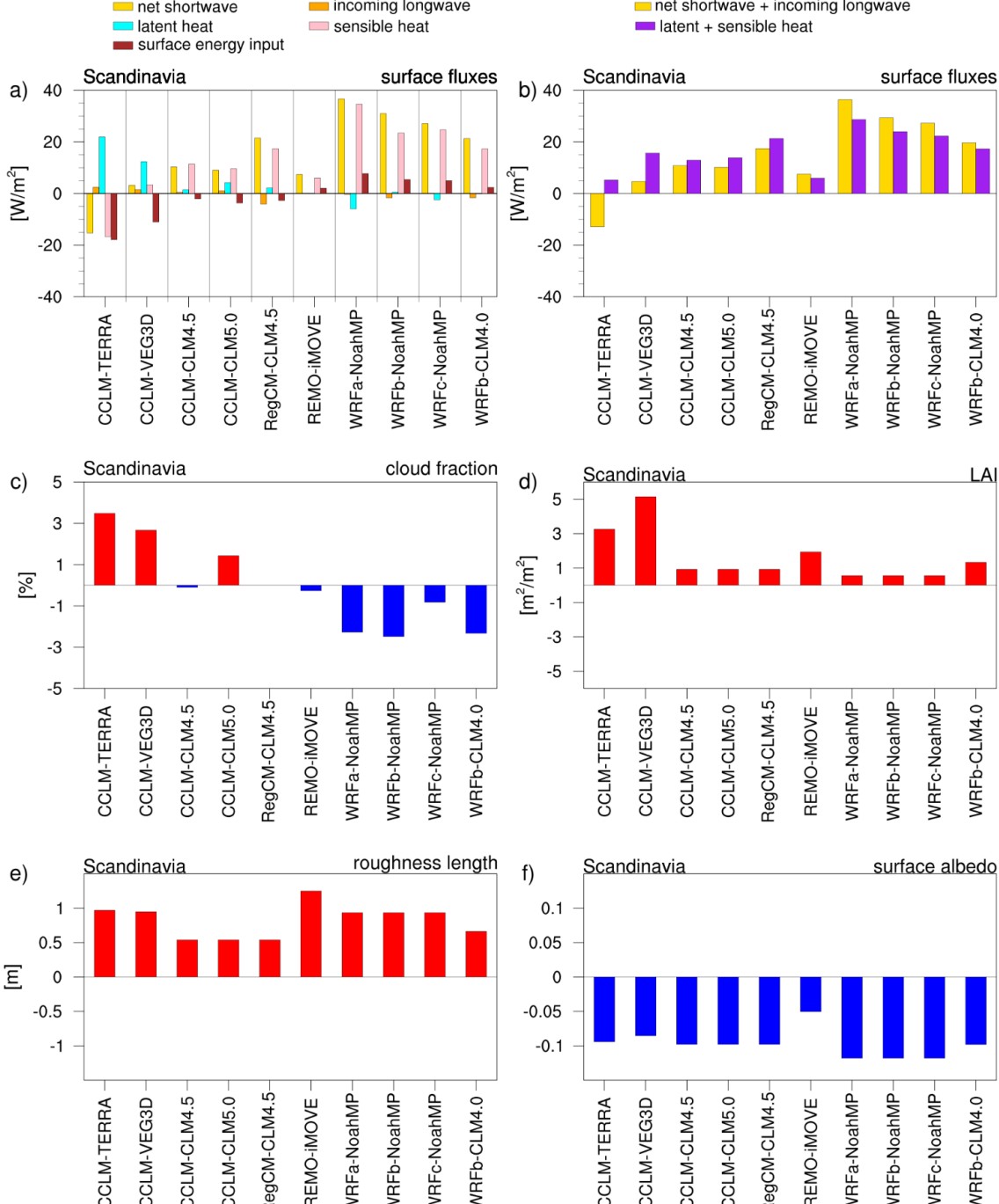

**Figure 7: (a)** Changes in surface energy balance components (FOREST minus GRASS) averaged over Scandinavia in summer, **(b)** The changes in available radiative energy at the surface and the sum of turbulent heat fluxes with afforestation (FOREST minus GRASS), **(c)** Cloud fraction response to afforestation across models, and the inter-model differences in leaf area index (LAI) **(d)**, surface roughness **(e)** and surface albedo **(f)** in summer (yearly maximum). Positive (negative) values mean increase (decrease) with afforestation.

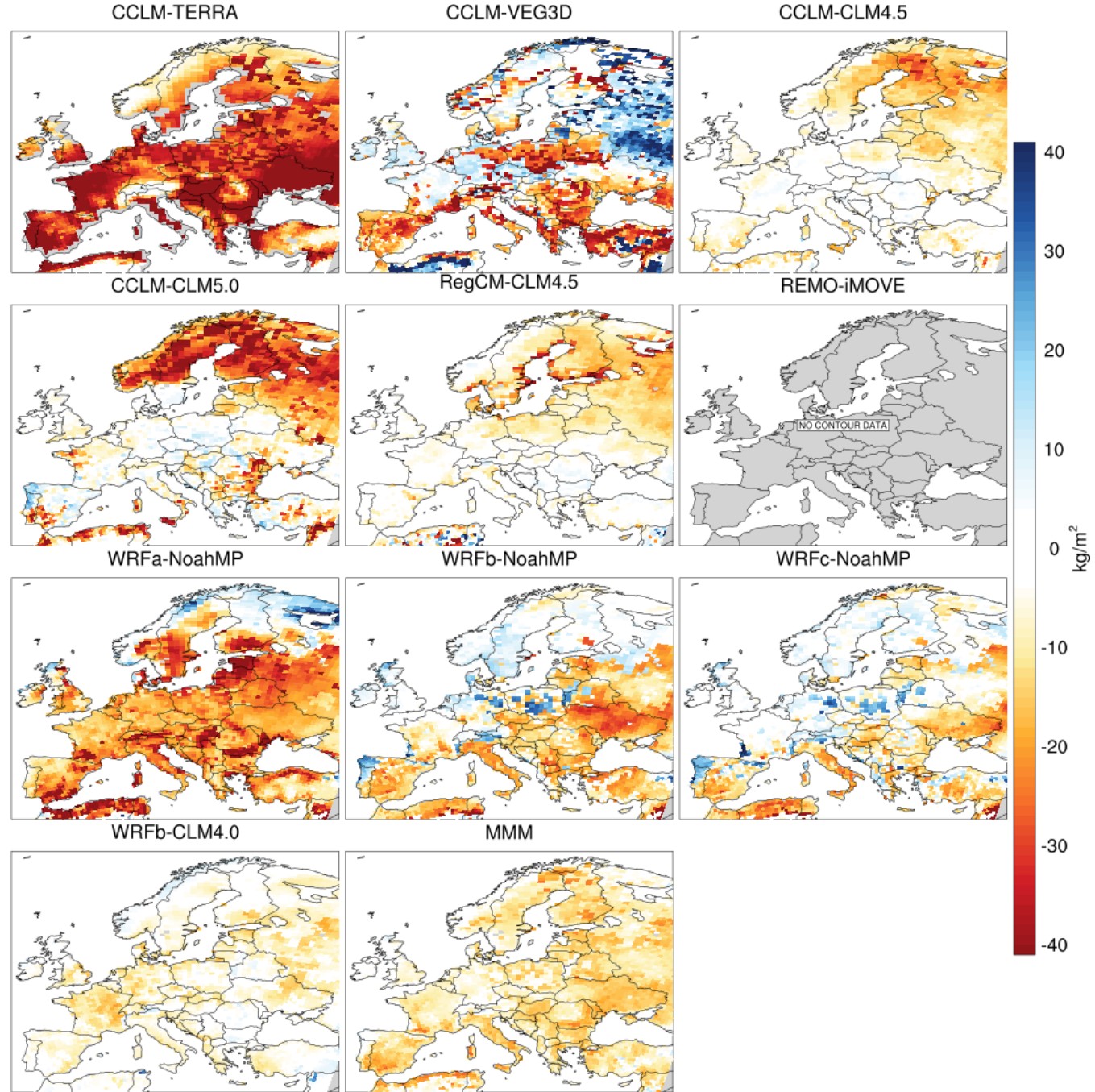

**Figure 8: Afforestation (FOREST minus GRASS) impact on soil moisture content (kg m⁻²) of the top 1 meter of the soil during**
**summer. REMO-iMOVE is not included because it employed a bucket scheme for soil hydrology in the LUCAS Phase 1 experiments, which does not allow a separation of soil moisture into different layers. Positive (negative) values mean an increase (decrease) due to afforestation.**

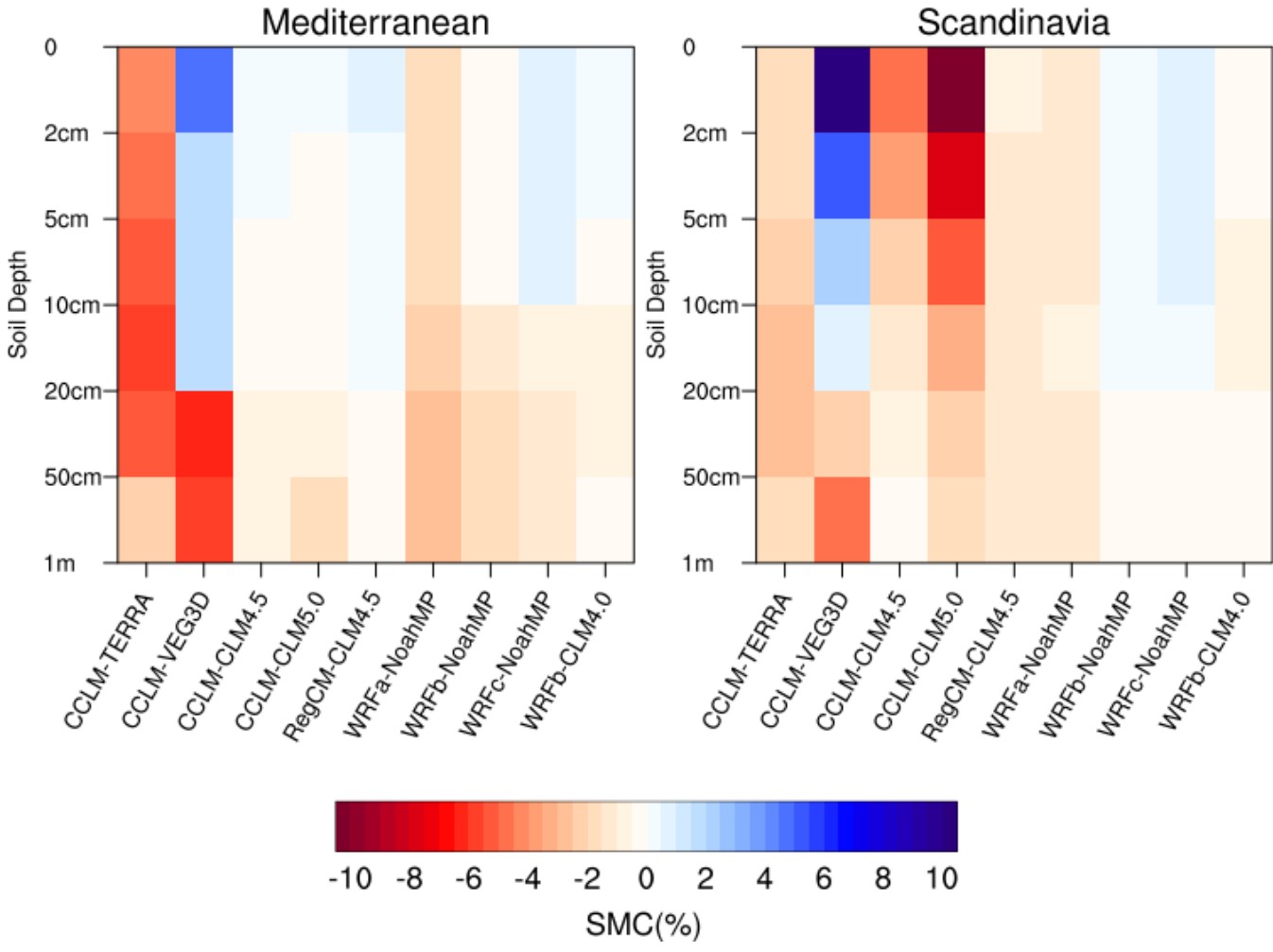

**Figure 9: Mean summer changes in soil moisture content (SMC) due to afforestation (FOREST minus GRASS) in the top 1 meter of the soil over Mediterranean and Scandinavia. Positive (negative) values mean an increase (decrease) due to afforestation.**

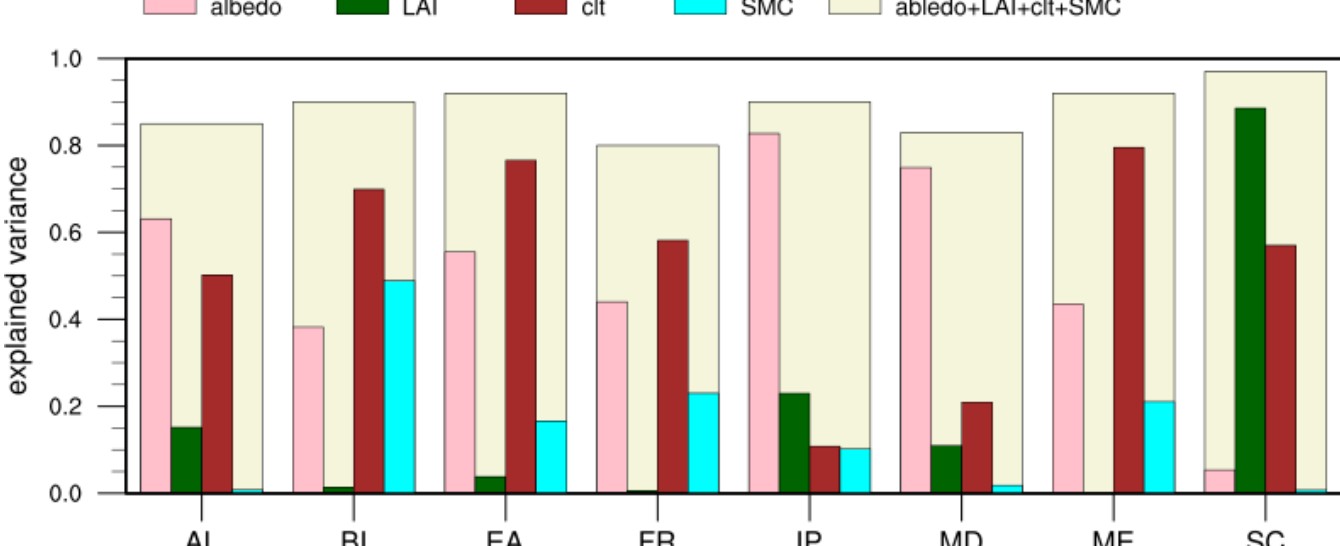

**Figure 10: The fraction of inter-model variance in AAST response (FOREST minus GRASS) explained by mean summer changes in albedo, leaf area index (LAI), cloud fraction (clt), soil moisture content (SMC) or all combined (albedo+LAI+clt+SMC). Bars represent the coefficient of determination ($R^2$) values derived from linear regression analysis applied over each sub-region. Alps (AL), British Isles (BI), Eastern Europe (EA), France (FR), Iberian Peninsula (IP), Mediterranean (MD), Mid-Europe (ME), Scandinavia (SC).**

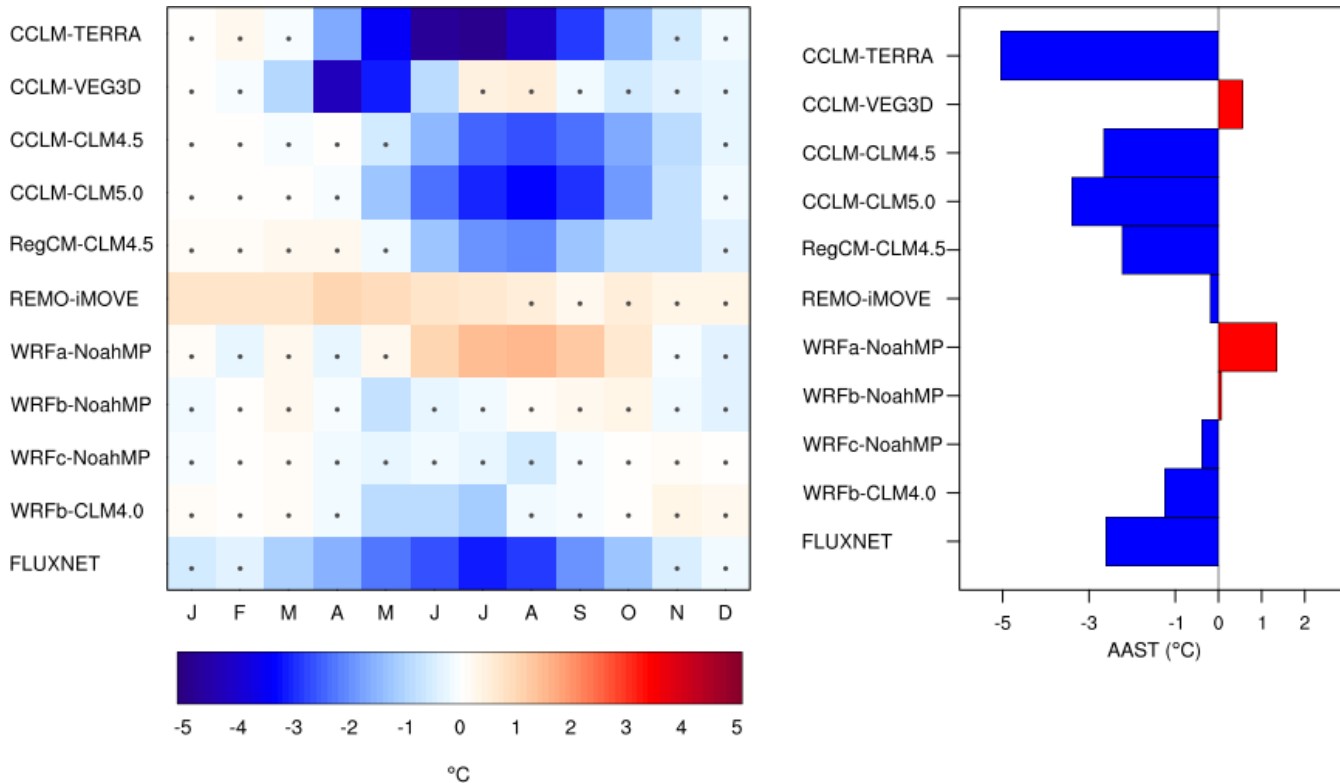

**Figure 11: Left:** Observed and simulated impact of afforestation on mean monthly soil temperature. The dots indicate the differences which are insignificantly different from zero in a two-sided t-test at 95% confidence level. **Right:** The changes in AAST (ºC) due to afforestation across models and observations. The observational differences are averaged over all the paired FLUXNET sites (forest minus open land) and the simulated changes are averaged over the corresponding model grids (FOREST minus GRASS). Positive (negative) values mean an increase (decrease) with afforestation.

755

760