# Peer review of "Afforestation impact on soil temperature in regional climate model simulations over Europe"

_Geoscientific Model Development, 2021_

## Author Comment (AC1)

We would like to thank the anonymous reviewers for their comprehensive comments. Below, we provide our responses in detail and describe the corresponding changes in the manuscript.

**Referee #1**

*RC1*: *(1) The paper lacks a clear causal explanation of why the models vary so much in the change in the amplitude of soil temperature. The attempt to explain the variation relies of two factors: the annual amplitude of ground heat flux, and soil moisture. Soil moistures is a perfectly valid explanatory variable but ground heat flux is not. Temperature and ground heat flux are both thermodynamic quantities and thus are very closely linked. Without an internal heat source (such are waste heat from soil carbon decay) subsurface temperature is surface heat flux modified changes in thermal diffusivity and heat capacity. In models thermal diffusivity is likely only being changed by soil moisture and maybe soil carbon content. Thus, it is no surprise that temperature and heat flux correlate well, but also this does not constitute an explanation.*

*Instead the focus should be on the differences in surface energy balance components (which are briefly examined) and the differences in model structure that may cause these differences. Key features to examine are: how snow is treated, how litter is treated (it is a good insulator), how forest canopies are treated and how root-depth is treated.*

**AC:** We agree that we should examine in-depth the reasons behind the changes in soil heating with afforestation and consequently in annual amplitude of soil temperature across modelling systems. We have noted that the changes in the annual amplitude of soil temperature are due to the representation of summertime climate processes, thus we should focus on summer season. Specifically, we will examine the differences in surface energy balance components across models and regions. The physical processes which take place at land surface, such as the radiative and turbulent heat fluxes, are differently weighted in models depending on land-use characteristics, like surface roughness, LAI, surface albedo etc. Analyzing the changes in surface energy budget components with respect to inter-model differences in land-use parameters, we will potentially reveal a large spread in the magnitude of afforestation effect on radiative and non-radiative processes and consequently in soil heating across models. Since the analysis is going to be carried out for summer, the land-surface is assumed to be snow-free. Heat storage in biomass or litter above ground is not considered in our models.

**Changes to manuscript**: We will change the section 3.2 (Annual amplitude of GHF) with the section "Surface energy balance" where an analysis focused on the summer changes of surface energy balance components will be performed, taking into account inter-model differences in surface albedo, LAI, cloud fraction, roughness length, which directly affect the energy and heat fluxes at land surface. It is expected that many of changes in land-use parameters will finally explain a large part of inter-model variance in AAST.

*RC1*: *(2) Despite being mentioned in the introduction soil profiles are never examined. Instead annual amplitudes of temperature at just one depth are examined. It would be useful to examine how temperature changed with depth in grassland and afforested conditions. Examining these profiles may also be helpful in finding a causal explanation for inter-model variance.*

**AC:** We had examined the soil temperature changes with depth, specifically we showed the simulated changes in soil temperature profile across seasons in Figures S9-S16 in the supplementary material. Although, you are right, we never mentioned any clear conclusion from these plots. Finally, the sign of temperature changes does not change with depth and only the magnitude of changes is affected.

**Changes to manuscript**: To better illustrate the changes in soil temperature with depth, we will add three additional figures, similar to Figure 2, where the AAST responses at 2 cm, 20 cm, 50 cm below the ground are to be shown (in addition to AAST response at 1 meter depth). Furthermore, we will include the soil temperature changes at the above-mentioned soil depths in our analysis on the mean monthly soil temperature changes due to afforestation over the regions of interest (Figure 3, Figure 4).

*RC1 :(3) How the models are being forced is unclear. The text implied that RCMs are being used with interactive atmospheres but the methods section seems to imply the reanalysis data is being used to force the models. The methods may be trying to say the reanalysis is being used at the RCM boundaries but this is not at all clear.*

**AC:** RCMs are forced by ERA-Interim reanalysis data at their lateral boundaries and at the lower boundary over sea.

**Changes to manuscript:** In line 115 we will add "..forced by ERA-Interim reanalysis data (Dee et al., 2011) at their lateral boundaries and at the lower boundary over sea".

*RC1 :(4) The manuscript has far to many abbreviations. As a rule of thumb, only define an abbreviation if you are going to use it 5 times or more.*

**AC:** Agreed

**Changes to manuscript**: We will reduce the abbreviations as much as possible.

*RC1 :(5) Citation parenthesis are used incorrectly. Citations are not placed in parenthesis if they need to be pronounced as part of a sentence. For example "(Davin and de Noblet-Ducoudre, 2010) analysed a GCM's sensitivity to idealized global deforestation ..." should be: "Davin and de Noblet-Ducoudre 2010, analysed a GCM's sensitivity to idealized global deforestation ..."*

**AC:** Agreed

**Changes to manuscript:** Citation parentheses will be used correctly

*RC1 :(6) Using Celsius instead of Kelvin would make the manuscript more readable.*

**AC:** Agreed

**Changes to manuscript**: Temperature unit will be converted to Celsius

**RC1** :*(7) The paper is not self-contained and relies on Davin et al. 2020. Elements critical for understanding the experiments should be reproduced here.*

**AC:** Agreed

**Changes to manuscript:** Vegetation maps and the table about RCMs characteristics and settings will be added in supplementary material.

*Specific Comments:*

**RC1:** *Abstract: Make it clearer you are examining soils.*

**AC***: Agreed*

**Changes to manuscript***: The abstract will be edited accordingly.*

**RC1:** *Introduction: Briefly introduce the biogeochemical effects of deforestation and make clear that you are only examining the biophysical effects. Also need to explain what RCMs are and how they improve on global studies.*

**AC***: Agreed.*

**Changes to manuscript***: We will add the proposed changes in the introduction section.*

**RC1:** *Line 44: Many of the models that you are referring to are Earth system models.*

**AC***: will be corrected*

**Changes to manuscript***: GCM will change to ESM*

**RC1:** *Line 48: Cloud feedbacks?*

**AC:** *maybe it's needed to make this sentence more readable.*

**Changes to manuscript***: Lines 46-50: "Davin and de Noblet-Ducoudre, 2010 analysed an ESM's sensitivity to idealized global deforestation, indicating that the net biophysical impact results from the balance between radiative and non-radiative processes. In the same study, deforestation induced a warming over the tropical zone owing to a reduction in evapotranspiration rate and surface roughness, whereas a deforestation-induced cooling simulated over the temperate and boreal zones, because an albedo increase provided the dominant influence in these regions. "*

**RC1:** *Line 48: "On contrary" is not grammatically correct. "However" would work.*

**AC:** Agreed

**Changes to manuscript***: will be corrected*

*RC1: Line 50: Citation needed.*

**AC***: maybe it's needed to make this sentence more readable.*

**Changes to manuscript***: Lines 46-50: "Davin and de Noblet-Ducoudre, 2010 analysed an ESM's sensitivity to idealized global deforestation, indicating that the net biophysical impact results from the balance between radiative and non-radiative processes. In the same study, deforestation induced a warming over the tropical zone owing to a reduction in evapotranspiration rate and surface roughness, whereas a deforestation-induced cooling simulated over the temperate and boreal zones, because an albedo increase provided the dominant influence in these regions. "*

*RC1: Line 54: "Inter-comparison" should be "Intercomparison"*

**AC:** Agreed

**Changes to manuscript***: will be corrected*

*RC1: Line 106: This table should be reproduced for this paper.*

**AC:** Agreed

**Changes to manuscript***: will be added in supplementary material*

*RC1: Line 110, 112: Forest and Grass are not acronyms and thus do not need to be in all caps.*

**AC***:* We would prefer to keep FOREST and GRASS in caps, as they are presented in previous studies of LUCAS FPS (Davin et al 2020, Breil et al 2020). They may not be acronyms, but they indicate the names of the experiments/scenarios under consideration. If they are written in lower case, there will be confusion between the names of the experiments and the general meaning of words "forest" and "grass".

**Changes to manuscript**: None

*RC1: Line 113: Show the maps.*

**AC***: Agreed*

**Changes to manuscript***:* Will be added in the supplementary material

*RC1: Line 117, 118: These abbreviations are barely used. They can easily be eliminated.*

**AC**: *Agreed*

**Changes to manuscript**: *will be removed*

**RC1:** *Figure 1: The map needs a North arrow, a scale, and inset showing the study domain, and higher resolution territorial boundaries. Using a different line style for national boarders and coastlines would also be helpful.*

**AC**: *Agreed*

**Changes to manuscript**: *The map will be refined.*

**RC1:** *Line 165-166: Rewrite sentence for clarity.*

**AC**: *Agreed*

**Changes to manuscript**: *In lines 165-166: "The theoretical maximum afforestation in RCMs has the potential to induce changes in large-scale atmospheric circulation, which can create teleconnections (Swann et al., 2012) that modify the regional cloud cover (Laguë and Swann, 2016) and thus the regional climate conditions. Such feedbacks are not realistic in observations, where most forest measurement locations are located in relatively small forest patches surrounded by open land and is almost unlikely to alter the climate conditions on regional scale."*

**RC1:** *178: 'assumed' is a poor choice of words. Models suggest.*

**AC**: *Agreed*

**Changes to manuscript**: *"assumed" will be changed to "suggested".*

**RC1:** *Line 207: Change 'involve' to 'include'*

**AC**: *Agreed*

**Changes to manuscript**: *"involve" will be changed to "include".*

**RC1:** *Line 213, 215: 'Obviously' and 'totally different' are informal constructions 'Clearly' and 'largely different' would be more consistent with formal English.*

**AC**: *Agreed*

**RC1:** *Figure 3,4: Use Celsius, also in caption give depth of temperature.*

**AC**: *Agreed*

**RC1:** *Figure 5: In caption explains which direction of heat flow is considered positive.*

**AC***: Agreed*

**RC1:** *Figure 8: Net radiation and turbulent fluxes should have opposite signs, one is opposing the other. Is melt energy included in the latent heat flux?*

**AC***: In this plot, we see the afforestation effect (FOREST minus GRASS) on radiative and heat fluxes at surface, thus positive (negative) values mean an increase (decrease) with afforestation.*

**Changes to manuscript***: We will make it clear what the direction of changes mean.*

**RC1:** *Figure 11: Write out the region names.*

**AC***: Agreed*

**RC1:** *Line 371: At what depth is the cooling?*

**AC**: In Table 2, we concentrated the characteristics of the sites selected from FLUXNET2015 dataset. More specifically, we provided the common measurement depth below the ground surface that is available for each pair site. The range of depths varies from 5 cm to 15 cm, with the most common depth being 10 cm for most pair sites. As already mentioned in section 2.3, soil temperature from models was linearly interpolated to the common measurement depth that is available for each pair site and averaged over the time period 2003-2014 which covers the observational time span.

**Changes to manuscript**: None

**RC1:** *Line 389: This section is the Discussion and Conclusions.*

**AC***: it will be corrected*

**RC1:** *Line 439: 'Nowadays' is English slang, very informal.*

**AC***: it will be corrected*

**References:**

*Breil, M., Rechid, D., Davin, E. L., Noblet-Ducoudré, N. de, Katragkou, E., Cardoso, R. M., Hoffmann, P., Jach, L. L., Soares, P. M. M., Sofiadis, G., Strada, S., Strandberg, G., Tölle, M. H., and Warrach-Sagi, K.: The Opposing Effects of Reforestation and Afforestation on the Diurnal Temperature Cycle at the Surface and*

in the Lowest Atmospheric Model Level in the European Summer, 33, 9159–9179, https://doi.org/10.1175/JCLI-D-19-0624.1, 2020.

Davin, E. L., Rechid, Di., Breil, M., Cardoso, R. M., Coppola, E., Hoffmann, P., Jach, L. L., Katragkou, E., De Noblet-Ducoudré, N., Radtke, K., Raffa, M., Soares, P. M. M., Sofiadis, G., Strada, S., Strandberg, G., Tölle, M. H., Warrach-Sagi, K., and Wulfmeyer, V.: Biogeophysical impacts of forestation in Europe: First results from the LUCAS (Land Use and Climate across Scales) regional climate model intercomparison, https://doi.org/10.5194/esd-11-183-2020, 2020.

Laguë, M. M. and Swann, A. L. S.: Progressive Midlatitude Afforestation: Impacts on Clouds, Global Energy Transport, and Precipitation, 29, 5561–5573, https://doi.org/10.1175/JCLI-D-15-0748.1, 2016.

Swann, A. L. S., Fung, I. Y., and Chiang, J. C. H.: Mid-latitude afforestation shifts general circulation and tropical precipitation, Proceedings of the National Academy of Sciences, 109, 712–716, https://doi.org/10.1073/pnas.1116706108, 2012.

---

## Author Comment (AC2)

We would like to thank the anonymous reviewers for their comprehensive comments. Below, we provide our responses in detail and describe the corresponding changes in the manuscript.

**Referee #2**

*RC2: lines 79-84: a brief justification may be needed here on why this study focuses on the "soil temperature profile" (by looking at the 1 m below ground in section 3.1) and not , also, the uppermost soil layer (= surface) temperature which is ultimately connected to the radiative and heat fluxes that drive the overlying air temperature, the surface climate parameter of main interest.*

*AC: As reported in authors' response to RC1 comments, we will add the soil temperature response to afforestation at three additional soil depths in results section.*

*Changes to manuscript: We will add the soil temperature response to afforestation at depths of 2 cm (close to uppermost soil layer and surface temperature), 20 cm and 50 cm, in addition to 1 meter.*

*RC2: line 98: as opposed to which PBL scheme in WRFb-NoahMP?*

*Changes to manuscript: we will add in line 99 "..as opposed to MYNN Level 2.5 TKE in WRFb-NoahMP.."*

*RC2: lines 125-128: is thermal diffusivity κ (see below) parameterised in the RCMs land surface schemes (and therefore derives from moisture affecting heat capacity, as you mention) or it is taken as a constant from look-up tables?  Could this property be shown for each model (especially if the authors feel it would assist interpretation of results)? From textbooks (pages 397-398 of McIlveen (2010) or section VIII.B. Conduction of Heat in Soil in Hillel (2003)) the thermal diffusivity is defined as:*
*κ = k/(ρC) where k = thermal conductivity ρ = density C = specific heat capacity The authors may consider the information in the Chen and Kling (1996) for better introducing and perhaps diagnosing in future studies, the thermal diffusivity κ.*

*AC: Thermal diffusivity is time-dependent and is parameterized in LSMs depending on the soil type, soil composition (organic matter content, mineral components), bulk density and soil wetness. We are not able to provide this hard-coded quantity for each model, as it is not usually a model output variable. Although, in our experiments soil composition and soil types are unchanged between the two land-use change scenarios, and only changes in soil wetness could have impact on thermal diffusivity. Also, RCMs do not account for possible occurrence of heat sources or sinks (such as organic matter or carbon decomposition) in the realm where soil heat flow takes place. In this way, we use soil moisture response to afforestation as an implication of changes in thermal diffusivity.*

*Changes to manuscript: We will better introduce thermal diffusivity. We will also add a column in table 1, where we are going to provide the parameterization schemes used from each model for calculation of thermal conductivity and volumetric heat capacity.*

*RC2: lines 128-129:  the fact that "GHF is calculated as the residual of surface energy balance because*

*the actual GHF outputs were not available in most models" assumes that model surface energy budgets are balanced, something that it may not be the case for, e.g., WRF (section 3.3 in Constantinidou et al., 2020a)*

***AC****: We agree that this should be mentioned*

***Changes to manuscript****: In Line 128-129 we will add the phrase "we define as energy input into ground the residual energy amount resulting from available radiative energy (net shortwave + incoming longwave radiation) minus the sum of turbulent heat fluxes, without accounting for likely deviation of surface energy budget from assumed balance in models (Constantinidou et al. 2020) "*

***RC2****: lines 169-170: Would it be useful to also show (in the Supplementary), not only the forest minus grass effect on the "annual amplitude of soil temperature (AAST) at 1 meter below the ground surface" (as done here), but the absolute value of annual land surface (skin) temperature as well?*

***AC****: We do not think that an annual mean of surface temperature would add value in our results. The large inter-model spread in AAST originates from summer temperature differences, whereas winter soil temperature sensitivity to afforestation is pretty small.  Thus, we focus on summer season. Also, the surface temperature response is strongly based on the residual of surface energy balance and has already been examined in previous studies established in LUCAS FPS (Davin et al 2020, Breil et al 2020). Moreover, in our revised manuscript, the soil temperature response to afforestation at 2 cm below the ground (close to uppermost soil layer and surface) across seasons will be added.*

***RC2****: line 230: Regarding the afforestation response of GHF, "Scandinavia appears to be the most sensitive among the regions". Any reasons?*

***AC****: The intensified coupling between surface and atmosphere in Scandinavia is caused by two factors; first, in Scandinavia forests consist of needleleaf forests with higher surface roughness (mixing-facilitating characteristic which enhance the heat exchange) compared to broadleaf trees dominating in the rest regions. Second, strong reductions in cloud fraction are noted in many models over Scandinavia, with result to intensify the albedo effect with afforestation.*

***Changes to manuscript****: In our revised manuscript, we will highlight the above-mentioned factors which affect the land-atmosphere coupling with afforestation in Scandinavia.*

***RC2:*** *lines 425-427: Can you also connect the results with the overarching ambition expressed in line 65 to "better constrain and represent the LUC biophysical forcing in regional climate simulations over Europe"?*

***AC****: Our sentence had unclear meaning, please let us reform this phrase*

***Changes to manuscript****: in line 65, the sentence "The crucial need to better constrain and represent the LUC biophysical forcing in regional climate simulations over Europe" will change to "The crucial need for the assessment of LUC biophysical impacts on regional scale over Europe".*

*RC2: lines 431-432: these proposed evaluations should critically include the land surface temperature, as in Constantinidou et al. (2020b)*

*AC: Agreed*

**Changes to manuscript**: *we will include the proposed study.*

*Minor/Technical Comments*

*The English need to be checked again as there are a few grammatical error or suboptimal expressions (some of them are listed below).*

*RC2: line 48: more correctly "On the contrary"*
*AC: Agreed*

*RC2: line 85: "second heat conduction law" can be written, more neatly, as "Fourier's second law of heat conduction". Same in lines 120, 226, 289*
*AC: Agreed*

*RC2: line 122: in equation (1), strictly, the derivative symbols should be replaced with partial differentials*
*AC: Agreed*

*RC2: line 127: "is the only variable which influence" should be "is the only variable which influences"*
*AC: will be corrected*

*RC2: line 291: "since affecting" should be replaced with "since it affects"*

*AC: will be corrected*

*RC2: line 402: replace "conducted an approach of" with "employed"*

*AC: will be corrected*

**References:**

Breil, M., Rechid, D., Davin, E. L., Noblet-Ducoudré, N. de, Katragkou, E., Cardoso, R. M., Hoffmann, P., Jach, L. L., Soares, P. M. M., Sofiadis, G., Strada, S., Strandberg, G., Tölle, M. H., and Warrach-Sagi, K.: The Opposing Effects of Reforestation and Afforestation on the Diurnal Temperature Cycle at the Surface and in the Lowest Atmospheric Model Level in the European Summer, 33, 9159–9179, https://doi.org/10.1175/JCLI-D-19-0624.1, 2020.

Davin, E. L., Rechid, Di., Breil, M., Cardoso, R. M., Coppola, E., Hoffmann, P., Jach, L. L., Katragkou, E., De Noblet-Ducoudré, N., Radtke, K., Raffa, M., Soares, P. M. M., Sofiadis, G., Strada, S., Strandberg, G., Tölle, M. H., Warrach-Sagi, K., and Wulfmeyer, V.: Biogeophysical impacts of forestation in Europe: First results from the LUCAS (Land Use and Climate across Scales) regional climate model intercomparison, https://doi.org/10.5194/esd-11-183-2020, 2020.

---

## Author Response (AR1)

We would like to thank the anonymous reviewers for their comprehensive comments. Below, we provide our responses in detail and describe the corresponding changes in the manuscript.

**Referee #1**

*RC1: (1) The paper lacks a clear causal explanation of why the models vary so much in the change in the amplitude of soil temperature. The attempt to explain the variation relies of two factors: the annual amplitude of ground heat flux, and soil moisture. Soil moistures is a perfectly valid explanatory variable but ground heat flux is not. Temperature and ground heat flux are both thermodynamic quantities and thus are very closely linked. Without an internal heat source (such are waste heat from soil carbon decay) subsurface temperature is surface heat flux modified changes in thermal diffusivity and heat capacity. In models thermal diffusivity is likely only being changed by soil moisture and maybe soil carbon content. Thus, it is no surprise that temperature and heat flux correlate well, but also this does not constitute an explanation.*

*Instead the focus should be on the differences in surface energy balance components (which are briefly examined) and the differences in model structure that may cause these differences. Key features to examine are: how snow is treated, how litter is treated (it is a good insulator), how forest canopies are treated and how root-depth is treated.*

**AC:** We agree that we should examine in-depth the reasons behind the changes in soil heating with afforestation and consequently in annual amplitude of soil temperature across modelling systems. We have noted that the changes in the annual amplitude of soil temperature are due to the representation of summertime climate processes, thus we should focus on summer season. Specifically, we will examine the differences in surface energy balance components across models and regions. The physical processes which take place at land surface, such as the radiative and turbulent heat fluxes, are differently weighted in models depending on land-use characteristics, like surface roughness, LAI, surface albedo etc. Addressing the changes in surface energy balance components with respect to inter-model differences in land-use parameters, we reveal a large spread in the magnitude of afforestation effect on radiative and non-radiative processes and consequently in soil heating across models. Since the analysis is performed for summer, the land-surface is assumed to be snow-free. Heat storage in biomass or litter above ground is not considered in our models.

**Changes to manuscript**: We have changed the section "3.2 Annual amplitude of GHF" with a new section "3.2 Surface energy availability", where the above-mentioned analysis is carried out. Also, the section "3.4 Attributing the inter-model spread in AAST to AAGHF and SMC" has been reformed in new section "3.4 The origin of inter-model spread in AAST" where additional explanatory variables (albedo, LAI, cloud fraction) have been included in multiple linear regression analysis, predicting more than 80% of inter-model variance in AAST in all regions.

*RC1: (2) Despite being mentioned in the introduction soil profiles are never examined. Instead annual amplitudes of temperature at just one depth are examined. It would be useful to examine how temperature changed with depth in grassland and afforested conditions. Examining these profiles may also be helpful in finding a causal explanation for inter-model variance.*

**AC:** We had examined the soil temperature changes with depth, specifically we showed the simulated changes in soil temperature profile across seasons in Figures S9-S16 in the supplementary material. Although, you are right, we never mentioned any clear conclusion from these plots. Finally, the sign of temperature change does not change with depth and only the magnitude of changes differs with depth.

**Changes to manuscript**: To better illustrate the changes in soil temperature with depth, we added three additional figures (Figure S2, Figure S3, Figure S4), similar to Figure 2, where the AAST response at 2 cm, 20 cm, 50 cm soil depth is depicted (in addition to AAST response at 1 meter depth). Furthermore, in Figure 3 and Figure 4, we examine the afforestation impact on mean monthly soil temperature at soil depths of 2 cm, 20 cm, 50 cm and 1 meter. The results are discussed in section 3.1.

*RC1 :(3) How the models are being forced is unclear. The text implied that RCMs are being used with interactive atmospheres but the methods section seems to imply the reanalysis data is being used to force the models. The methods may be trying to say the reanalysis is being used at the RCM boundaries but this is not at all clear.*

**AC:** RCMs are forced by ERA-Interim reanalysis data at their lateral boundaries and at the lower boundary over sea.

**Changes to manuscript:** In line 121-122 we added "..forced by ERA-Interim reanalysis data (Dee et al., 2011) at their lateral boundaries and at the lower boundary over sea".

*RC1 :(4) The manuscript has far to many abbreviations. As a rule of thumb, only define an abbreviation if you are going to use it 5 times or more.*

**AC:** Agreed

**Changes to manuscript**: We have reduced the abbreviations as much as possible.

*RC1 :(5) Citation parenthesis are used incorrectly. Citations are not placed in parenthesis if they need to be pronounced as part of a sentence. For example "(Davin and de Noblet-Ducoudre, 2010) analysed a GCM's sensitivity to idealized global deforestation ..." should be: "Davin and de Noblet-Ducoudre 2010, analysed a GCM's sensitivity to idealized global deforestation ..."*

**AC:** Agreed

**Changes to manuscript:** Citation parentheses have been corrected.

*RC1 :(6) Using Celsius instead of Kelvin would make the manuscript more readable.*

**AC:** Agreed

**Changes to manuscript**: temperature unit has been changed to Celsius.

*RC1 :(7) The paper is not self-contained and relies on Davin et al. 2020. Elements critical for understanding the experiments should be reproduced here.*

**AC:** Agreed

**Changes to manuscript:** The vegetation maps used in FOREST and GRASS experiment are shown in Figure S1. The full table about RCMs characteristics and settings has been also added in supplementary material (Table S1).

*Specific Comments:*

**RC1:** *Abstract: Make it clearer you are examining soils.*

**AC**: *Agreed*

**Changes to manuscript**: *Abstract has been re-written accordingly.*

**RC1:** *Introduction: Briefly introduce the biogeochemical effects of deforestation and make clear that you are only examining the biophysical effects. Also need to explain what RCMs are and how they improve on global studies.*

**AC**: *Agreed.*

**Changes to manuscript**: In lines 40-42 we added *"*The biogeochemical effects of afforestation or reforestation are mostly related to increased carbon stocks stored in vegetation and soil, as the total carbon stored in forests is nearly three times larger than carbon stored in croplands (Devaraju et al., 2015)".

In line 60-63 we added "Regional Climate Models (RCMs) constitute dynamical downscaling techniques applied over limited-area domains with boundary conditions either from global reanalysis or global climate model (GCM) output (Katragkou et al., 2015; Giorgi, 2019; Rummukainen, 2016). RCMs operate on higher resolutions than GCMs adding value in regions with complex orography and capturing exreme events (Soares et al., 2012; Cardoso et al., 2013; Warrach-Sagi et al., 2013)"

*In line 64-65 we write "Here, we investigate the biophysical impact of afforestation on soil temperature across Europe.."*

**RC1:** *Line 44: Many of the models that you are referring to are Earth system models.*

**AC**: *Agreed*

**Changes to manuscript**: *"GCM" changed to "ESM" in line 47.*

**RC1:** *Line 48: Cloud feedbacks?*

**AC:** *maybe it's needed to make this sentence more readable.*

**Changes to manuscript**: *Lines 48-53: "Davin and de Noblet-Ducoudre, 2010 analyzed an ESM's sensitivity to idealized global deforestation, indicating that the net biophysical impact results from the balance between radiative and non-radiative processes. In the same study, deforestation induced a warming over the tropical zone owing to a reduction in evapotranspiration rate and surface roughness, whereas a deforestation-induced cooling simulated over the temperate and boreal zones, because an albedo increase provided the dominant influence in these regions."*

**RC1:** *Line 48: "On contrary" is not grammatically correct. "However" would work.*

**AC:** Agreed

**Changes to manuscript**: *the sentence has been reformed.*

**RC1:** *Line 50: Citation needed.*

**AC**: *maybe it's needed to make this sentence more readable.*

**Changes to manuscript**: *Lines 48-53: "Davin and de Noblet-Ducoudre, 2010 analyzed an ESM's sensitivity to idealized global deforestation, indicating that the net biophysical impact results from the balance between radiative and non-radiative processes. In the same study, deforestation induced a warming over the tropical zone owing to a reduction in evapotranspiration rate and surface roughness, whereas a deforestation-induced cooling simulated over the temperate and boreal zones, because an albedo increase provided the dominant influence in these regions."*

**RC1:** *Line 54: "Inter-comparison" should be "Intercomparison"*

**AC:** Agreed

**Changes to manuscript**: *has been corrected*

**RC1:** *Line 106: This table should be reproduced for this paper.*

**AC:** Agreed

**Changes to manuscript**: *has been added as Table S1 in the supplementary material.*

**RC1:** *Line 110, 112: Forest and Grass are not acronyms and thus do not need to be in all caps.*

**AC**: We would prefer to keep FOREST and GRASS in caps, as they are presented in previous studies of LUCAS FPS (Davin et al 2020, Breil et al 2020). They may not be acronyms, but they indicate the names of

the experiments/scenarios under consideration. If they are written in lower case, there will be confusion between the names of the experiments and the general meaning of words "forest" and "grass".

**Changes to manuscript**: None

*RC1: Line 113: Show the maps.*

**AC***: Agreed*

**Changes to manuscript***: have been added in Figure S1 in the supplementary material.

*RC1: Line 117, 118: These abbreviations are barely used. They can easily be eliminated.*

*AC: Agreed*

**Changes to manuscript***:  have been removed*

*RC1: Figure 1: The map needs a North arrow, a scale, and inset showing the study domain, and higher resolution territorial boundaries. Using a different line style for national boarders and coastlines would also be helpful.*

**AC***: Agreed*

**Changes to manuscript***: The plot (Figure 1) has been remapped.*

*RC1: Line 165-166: Rewrite sentence for clarity.*

*AC: Agreed*

**Changes to manuscript***: In lines 172-176: "The theoretical maximum afforestation in RCMs has the potential to induce changes in large-scale atmospheric circulation, which can create teleconnections (Swann et al., 2012) that modify the regional cloud cover (Laguë and Swann, 2016) and thus the regional climate conditions. Such feedbacks are not realistic in observations, where most forest measurement locations are located in relatively small forest patches surrounded by open land and is almost unlikely to alter the climate conditions on regional scale."*

*RC1: 178: 'assumed' is a poor choice of words. Models suggest.*

**AC***: Agreed*

**Changes to manuscript***: "assumed" changed to "suggested".*

*RC1: Line 207: Change 'involve' to 'include'*

**AC***: Agreed*

**Changes to manuscript***: "involve" changed to "include".*

*RC1: Line 213, 215: 'Obviously' and 'totally different' are informal constructions 'Clearly' and 'largely different' would be more consistent with formal English.*

**AC***: Agreed*

**Changes to manuscript***: the sentence has been reformed*

*RC1: Figure 3,4: Use Celsius, also in caption give depth of temperature.*

**AC***: Agreed*

**Changes to manuscript:** Celsius is used, soil depths have been added in figures 3, 4.

*RC1: Figure 5: In caption explains which direction of heat flow is considered positive.*

**AC***: In this plot, we see the afforestation effect (FOREST minus GRASS) on surface energy availability, thus positive (negative) values mean an increase (decrease) with afforestation.*

**Changes to manuscript:** In caption, we have added the phrase "Positive (negative) values mean an increase (decrease) with afforestation"

*RC1: Figure 8: Net radiation and turbulent fluxes should have opposite signs, one is opposing the other. Is melt energy included in the latent heat flux?*

**AC***: In this plot, we see the afforestation effect (FOREST minus GRASS) on radiative and heat fluxes at surface, thus positive (negative) values mean an increase (decrease) with afforestation.*

**Changes to manuscript***: In captions of figures 2,5,6,7,8,9,11 we have added the phrase "Positive (negative) values mean an increase (decrease) with afforestation"*

*RC1: Figure 11: Write out the region names.*

**AC***: Agreed*

**Changes to manuscript:** Regions names have been written out in Figure 9.

*RC1: Line 371: At what depth is the cooling?*

**AC**: In Table 2, we concentrated the characteristics of the sites selected from FLUXNET2015 dataset. More specifically, we provided the common measurement depth below the ground surface that is available for each pair site. The range of depths varies from 5 cm to 15 cm, with the most common depth being 10 cm for most pair sites. As already mentioned in section 2.3, soil temperature from models was linearly interpolated to the common measurement depth that is available for each pair site and averaged over the time period 2003-2014 which covers the observational time span.

**Changes to manuscript**: None

*RC1:* *Line 389: This section is the Discussion and Conclusions.*

**AC***: Agreed*

**Changes to manuscript:** Section 4 "Discussion and Conclusions"

*RC1:* *Line 439: 'Nowadays' is English slang, very informal.*

**AC***: Agreed*

**Changes to manuscript:** line 431**:** "nowadays" changed to "last years"

**Referee #2**

*RC2: lines 79-84: a brief justification may be needed here on why this study focuses on the "soil temperature profile" (by looking at the 1 m below ground in section 3.1) and not , also, the uppermost soil layer (= surface) temperature which is ultimately connected to the radiative and heat fluxes that drive the overlying air temperature, the surface climate parameter of main interest.*

**AC:** We had examined the soil temperature changes with depth, specifically we showed the simulated changes in soil temperature profile across seasons in Figures S9-S16 in the supplementary material. Although, you are right, we never mentioned any clear conclusion from these plots. Finally, the sign of temperature change does not change with depth and only the magnitude of changes differs with depth.

**Changes to manuscript**: To better illustrate the changes in soil temperature with depth, we added three additional figures (Figure S2, Figure S3, Figure S4), similar to Figure 2, where the AAST response at 2 cm (close to surface and uppermost soil layer), 20 cm, 50 cm soil depth is depicted (in addition to AAST response at 1 meter depth). Furthermore, in Figure 3 and Figure 4, we examine the afforestation impact on mean monthly soil temperature at soil depths of 2 cm, 20 cm, 50 cm and 1 meter. The results are discussed in section 3.1.

*RC2: line 98: as opposed to which PBL scheme in WRFb-NoahMP?*

**Changes to manuscript***: we added in line 111 "..as opposed to MYNN Level 2.5 TKE in WRFb-NoahMP.."*

**RC2:** *lines 125-128: is thermal diffusivity κ (see below) parameterised in the RCMs land surface schemes (and therefore derives from moisture affecting heat capacity, as you mention) or it is taken as a constant from look-up tables? Could this property be shown for each model (especially if the authors feel it would assist interpretation of results)? From textbooks (pages 397-398 of McIlveen (2010) or section VIII.B. Conduction of Heat in Soil in Hillel (2003)) the thermal diffusivity is defined as:*
*κ = k/(ρC) where k = thermal conductivity ρ = density C = specific heat capacity The authors may consider the information in the Chen and Kling (1996) for better introducing and perhaps diagnosing in future studies, the thermal diffusivity κ.*

**AC**: *Thermal diffusivity is time-dependent and is parameterized in LSMs depending on the soil type, soil composition (organic matter content, mineral components), bulk density and soil wetness. We are not able to provide this hard-coded quantity for each model, as it is not usually a model output variable. Although, in our experiments soil composition and soil types are unchanged between the two land-use change scenarios, and only changes in soil wetness could have impact on thermal diffusivity. Also, RCMs do not account for possible occurrence of heat sources or sinks (such as organic matter or carbon decomposition) in the realm where soil heat flow takes place. In this way, we use soil moisture response to afforestation as an implication of changes in thermal diffusivity.*

**Changes to manuscript**: *In lines 132-137 we write "In RCMs, k is time dependent and is parameterized depending on soil type and composition (mineral components, organic matter content), on bulk density and soil wetness. In our experiments, soil texture remains unchanged and RCMs do not account for possible occurrence of heat sources or sinks (such as organic matter or carbon decomposition) in the realm where soil heat flow takes place. Thus, the potential changes in soil wetness with afforestation constitute the main driver of differences in soil thermal diffusivity in our experiments. In this way, we use soil moisture response to afforestation as a potentially explanatory variable of soil temperature variations in RCMs."*

*Also, in Table 1 in column "Soil column" we have added the parameterizations schemes used by each model for calculation of thermal conductivity and volumetric heat capacity.*

**RC2**: *lines 128-129: the fact that "GHF is calculated as the residual of surface energy balance because the actual GHF outputs were not available in most models" assumes that model surface energy budgets are balanced, something that it may not be the case for, e.g., WRF (section 3.3 in Constantinidou et al., 2020a)*
**AC**: *We agree that this should be mentioned.*

**Changes to manuscript**: *In Lines 139-142 we added the phrase "we define as energy input into the ground the residual energy amount resulting from available radiative energy (net shortwave + incoming longwave radiation) minus the sum of turbulent heat fluxes, without accounting for likely deviation of surface energy budget from assumed balance in models (Constantinidou et al. 2020b) "*

*RC2*: lines 169-170: Would it be useful to also show (in the Supplementary), not only the forest minus grass effect on the "annual amplitude of soil temperature (AAST) at 1 meter below the ground surface" (as done here), but the absolute value of annual land surface (skin) temperature as well?

*AC*: We do not think that an annual mean of surface temperature would add value in our results. The large inter-model spread in AAST originates from summer temperature differences, whereas winter soil temperature sensitivity to afforestation is pretty small. Thus, we focus on summer season. Also, the surface temperature response is strongly based on the residual of surface energy balance and has already been examined in previous studies established in LUCAS FPS (Davin et al 2020, Breil et al 2020). Moreover, in our revised manuscript, the soil temperature response to afforestation at 2 cm below the ground (close to uppermost soil layer and surface) across seasons has been added.

*RC2*: line 230: Regarding the afforestation response of GHF, "Scandinavia appears to be the most sensitive among the regions". Any reasons?

*AC*: The intensified coupling between surface and atmosphere in Scandinavia is caused by two factors; first, in Scandinavia forests consist of needleleaf forests with higher surface roughness (mixing-facilitating characteristic which enhance the heat exchange) compared to broadleaf trees dominating in the rest regions. Second, strong reductions in cloud fraction are noted in many models over Scandinavia, with result to intensify the albedo effect with afforestation.

*Changes to manuscript*: In our revised manuscript in lines 251-272, we highlight the above-mentioned factors which affect the land-atmosphere coupling with afforestation in Scandinavia.

*RC2:* lines 425-427: Can you also connect the results with the overarching ambition expressed in line 65 to "better constrain and represent the LUC biophysical forcing in regional climate simulations over Europe"?

*AC*: Our sentence had unclear meaning, please let us reform this phrase.

*Changes to manuscript*: in line 65, the sentence "The crucial need to better constrain and represent the LUC biophysical forcing in regional climate simulations over Europe" changed to "The crucial need for the assessment of LUC biophysical impacts on regional scale over Europe..".

*RC2*: lines 431-432: these proposed evaluations should critically include the land surface temperature, as in Constantinidou et al. (2020b)

*AC*: Agreed

*Changes to manuscript*: in lines 422-424 we write "Future studies should focus on the evaluation of model performances, similar to Katragkou et al., 2015 and Constantinidou et al., 2020a, in order to identify the origins of systematic biases and improve the representation of climate processes in simulations"

*Minor/Technical Comments*

*The English need to be checked again as there are a few grammatical error or suboptimal expressions (some of them are listed below).*

**RC2**: *line 48: more correctly "On the contrary"*
**AC**: *Agreed*

**Changes to manuscript**: *has been corrected*

**RC2**: *line 85: "second heat conduction law" can be written, more neatly, as "Fourier's second law of heat conduction". Same in lines 120, 226, 289*
**AC**: *Agreed*

**Changes to manuscript**: *has been corrected*

**RC2**: *line 122: in equation (1), strictly, the derivative symbols should be replaced with partial differentials*
**AC**: *Agreed*

**Changes to manuscript**: *has been corrected*

**RC2:** *line 127: "is the only variable which influence" should be "is the only variable which influences"*
**AC**: *Agreed*

**Changes to manuscript**: *has been corrected*

**RC2**: *line 291: "since affecting" should be replaced with "since it affects"*

**AC**: *Agreed*

**Changes to manuscript**: *has been corrected*

**RC2**: *line 402: replace "conducted an approach of" with "employed"*

**AC**: *Agreed*

**Changes to manuscript**: *has been corrected*

**References:**

Breil, M., Rechid, D., Davin, E. L., Noblet-Ducoudré, N. de, Katragkou, E., Cardoso, R. M., Hoffmann, P., Jach, L. L., Soares, P. M. M., Sofiadis, G., Strada, S., Strandberg, G., Tölle, M. H., and Warrach-Sagi, K.: The Opposing Effects of Reforestation and Afforestation on the Diurnal Temperature Cycle at the Surface and in the Lowest Atmospheric Model Level in the European Summer, 33, 9159–9179, https://doi.org/10.1175/JCLI-D-19-0624.1, 2020.

Davin, E. L., Rechid, Di., Breil, M., Cardoso, R. M., Coppola, E., Hoffmann, P., Jach, L. L., Katragkou, E., De Noblet-Ducoudré, N., Radtke, K., Raffa, M., Soares, P. M. M., Sofiadis, G., Strada, S., Strandberg, G., Tölle, M. H., Warrach-Sagi, K., and Wulfmeyer, V.: Biogeophysical impacts of forestation in Europe: First results from the LUCAS (Land Use and Climate across Scales) regional climate model intercomparison, https://doi.org/10.5194/esd-11-183-2020, 2020.

Laguë, M. M. and Swann, A. L. S.: Progressive Midlatitude Afforestation: Impacts on Clouds, Global Energy Transport, and Precipitation, 29, 5561–5573, https://doi.org/10.1175/JCLI-D-15-0748.1, 2016.

Swann, A. L. S., Fung, I. Y., and Chiang, J. C. H.: Mid-latitude afforestation shifts general circulation and tropical precipitation, Proceedings of the National Academy of Sciences, 109, 712–716, https://doi.org/10.1073/pnas.1116706108, 2012.